# Single-molecule dynamics reveal ATP binding alone powers substrate translocation by an ABC transporter

Christoph Nocker [●], Matija Pečak, Tobias Nocker [●], Amin Fahim [●],
Lukas Sušac & Robert Tampé [●] [✉]

ATP-binding cassette (ABC) transporters are molecular machines involved in diverse physiological processes, including antigen processing by TAP, a key component of adaptive immunity. TAP and its bacterial homolog TmrAB use ATP to translocate peptides across membranes, yet the precise mechanism linking ATP binding to substrate movement remains unclear. Here, we employ a single-molecule FRET sensor to visualize single translocation events by individual ABC transporters and thereby overcome the limitations of ensemble averaging. This approach reveals that substrate transport is driven by a conformational switch from the inward- to the outward-facing state. Using a slow-turnover TmrAB variant, we demonstrate that ATP binding alone, even in the absence of $Mg^{2+}$, is sufficient to drive a single round of peptide translocation. Cryo-EM structures of wild-type and slow-turnover TmrAB show that ATP binding induces the outward-facing conformation even without $Mg^{2+}$. In wild-type TmrAB, this conformational transition supports a single translocation event, whereas $Mg^{2+}$-dependent ATP hydrolysis is required to reset the transporter. These findings establish a direct mechanistic link between ATP binding and substrate translocation at single-molecule resolution and provide insight into the catalytic cycle of ABC transporters.

ABC transporters represent the largest family of primary active transport proteins, conserved across all domains of life[1-3]. Using the energy from ATP binding and hydrolysis, they move a broad spectrum of substrates, e.g., ions, amino acids, lipids, vitamins, proteins, and peptides, against concentration gradients across biological membranes. Because of their central roles in human diseases, including multidrug resistance, bacterial pathogenicity, and genetic disorders, a detailed understanding of their molecular mechanisms is of significant clinical importance[4-6].

ABC proteins are classified according to conserved sequence features within their nucleotide-binding domains (NBDs)[7] and structural organization of their transmembrane domains (TMDs)[8]. Among type IV ABC transporters, P-glycoprotein (P-gp, MDR1) and heterodimeric transporters such as the transporter associated with antigen processing (TAP1/2) and its prokaryotic homolog, the *Thermus*

*thermophilus* multidrug resistance protein A and B (TmrAB), are notable examples[8-10]. TAP plays a pivotal role in adaptive immunity by transporting antigenic peptides from the cytosol into the endoplasmic reticulum, where they are loaded onto major histocompatibility complex (MHC) class I molecules for immune surveillance[11,12]. TmrAB, a structural and functional homolog of TAP with overlapping substrate specificity, can rescue TAP deficiency in human cells[13]. Cryogenic electron microscopy (cryo-EM) analyses have captured multiple conformational states of TmrAB during turnover[14]. However, ensemble approaches cannot resolve the order or kinetics of individual substrate-translocation steps, leaving key mechanistic questions open.

A central challenge in understanding ABC transporter energetics is disentangling the contributions of ATP binding, ATP hydrolysis, and the involvement of $Mg^{2+}$ as a cofactor. It remains unclear whether ATP binding alone is sufficient to initiate substrate translocation or whether

---

Institute of Biochemistry, Biocenter, Goethe University Frankfurt, Frankfurt am Main, Germany. [✉]e-mail: tampe@em.uni-frankfurt.de

hydrolysis is also required[15,16]. The role of $Mg^{2+}$ remains controversial: while some studies suggest it is essential for ATP binding[17], others indicate that NBD dimerization can occur in its absence[18–20]. We hypothesized that ATP binding alone drives single-substrate translocation, whereas $Mg^{2+}$-dependent ATP hydrolysis is necessary to reset the transporter from the outward-facing (OF) to the inward-facing (IF) state.

Heterodimeric type IV ABC transporters, including TAP1/2 and TmrAB, add another level of complexity due to their intrinsic functional asymmetry[9,21–23]. This asymmetry complicates mechanistic interpretation, particularly with respect to the efficiency and modulation of the transport cycle—areas that are not yet fully understood. Resolving these issues requires approaches that go beyond structural snapshots and ensemble biochemical assays, enabling direct observation of individual transport events.

Single-molecule techniques overcome ensemble averaging and provide high-resolution insights into transport dynamics[24–26]. In particular, single-molecule Förster resonance energy transfer (smFRET) studies confirmed an alternating-access mechanism in ABC transporters; for example, the bacterial homodimeric exporter McjD requires both substrate and ATP to adopt the OF conformation[27]. Moreover, combining smFRET with single-particle cryo-EM enabled visualization of the ABC transporter MRP1 under turnover conditions[28]. Conformational changes in substrate-binding proteins (SBP) from bacterial ABC importers were investigated by smFRET[29,30]. The SBPs were repurposed as smFRET sensors to track individual translocation events in secondary active transporters, providing quantitative amino acid transport rates[31,32]. A related sensor derived from the substrate-binding protein OppA from *Lactococcus lactis* (*Ll*OppA) was also used to characterize substrate binding at the single-molecule level[30].

Here, we establish a single-molecule assay that directly visualizes peptide transport by an individual TAP-related heterodimeric ABC transporter. Using an *Ll*OppA-based smFRET sensor encapsulated within liposomes, we monitor substrate translocation by TmrAB at single-molecule resolution. Our results reveal that ATP binding alone is sufficient to induce a conformational switch that drives peptide transport in the absence of $Mg^{2+}$, whereas $Mg^{2+}$-dependent ATP hydrolysis is required to reset the transporter and complete the transport cycle.

## Results

### Establishing an smFRET sensor for antigenic peptides

To detect antigenic peptides of diverse sequences and lengths, similar to those presented on MHC I molecules in adaptive immunity, we exploited the binding promiscuity of a bacterial substrate-binding protein. We used a double-cysteine variant of *Ll*OppA (*Ll*OppA$^{A209C/S441C}$)[30], which was site-specifically labeled at the engineered cysteine residues with the FRET pair Alexa Fluor 555 (AF555) and AF647 (hereafter referred to as OppA) (Supplementary Fig. 1). Binding affinities of the model antigenic peptide RRYQKSTEL to unlabeled and labeled OppA were determined using three independent biophysical approaches: tryptophan fluorescence quenching, ensemble FRET, and single-molecule FRET. All yielded consistent dissociation constants $K_D$ ranging from 0.2 to 0.9 μM (Fig. 1d, Supplementary Fig. 1c, e).

To characterize the FRET sensor at the single-molecule level, OppA was immobilized on a PEGylated glass surface using a biotinylated anti-His antibody and streptavidin, with biotin-PEG included at low density for specific tethering. Individual OppA molecules were imaged using total-internal reflection fluorescence (TIRF) microscopy with alternating laser excitation (ALEX) (Fig. 1a, b, Supplementary Fig. 2). In the absence of peptides (apo state), OppA adopted a low FRET efficiency ($E = 0.6$), which was shifted to a high-FRET state ($E = 0.9$) upon addition of increasing peptide concentrations (0–100 μM) (Fig. 1c, Supplementary Figs. 3 and 4). At peptide concentrations near the $K_D$ value, OppA exhibited dynamic transitions between low- and high-FRET states, indicating OppA-peptide association and dissociation events (Fig. 1c). Hidden Markov Modeling (HMM) showed concentration-dependent dwell times derived from dynamic smFRET traces (Fig. 1e). The closing rates, ranging from 1.6 to 4.5 s$^{-1}$, increased linearly with peptide concentrations between 1 and 10 μM ($k_{on} = 0.29 \pm 0.02$ μM$^{-1}$s$^{-1}$ from $n = 1311$ single-molecule traces) (Fig. 1f), validating the sensor's capability to monitor peptide concentrations via binding kinetics.

### Rationale for single-substrate sensors in liposomes

For single-molecule transport measurements, we employed both wild-type TmrAB (TmrAB$^{WT}$) and a slow-turnover variant generated by substituting the catalytic glutamate with glutamine (TmrA$^{E523Q}$B, hereafter TmrA$^{EQ}$B), which exhibits ~1000-fold reduced ATP hydrolysis[13]. Both variants were expressed and purified as previously described[14] (Supplementary Fig. 5a, b). To track the translocation of individual peptides, liposomes of defined size were required to generate lumenal peptide concentrations within the linear detection range of OppA closing rates. Liposomes with a diameter of 100 nm correspond to an internal volume of ~0.5 aL, yielding an effective peptide concentration of ~3.2 μM per translocated peptide. This concentration aligns with the $K_D$ value and linear detection range for the closing rates measured by the smFRET sensor (Fig. 1f). To statistically avoid populations of liposomes containing multiple smFRET sensors or transporters, we used a substoichiometric encapsulation/reconstitution ratio of 0.2 OppA and 0.1 TmrAB per liposome (Supplementary Fig. 6a–c).

To selectively capture single uptake-competent transport complexes, we used a conformation-independent nanobody (Nb9F10)[14] tethered to the surface via PEG$_{11}$-biotin and streptavidin (Supplementary Fig. 5c–e). This strategy selectively retained liposomes with correctly oriented, uptake-competent transporters while removing empty liposomes and liposomes with misoriented transporters. Immobilization specificity was confirmed with and without nanobody tethering (Supplementary Fig. 6d). Flow cytometry-based single-liposome assays verified that nanobody binding did not alter transport activity (Supplementary Fig. 7).

Sensor functionality after liposome encapsulation was confirmed by measuring the fluorescence lifetime and anisotropy of the dyes, which showed no membrane-induced perturbations (Supplementary Fig. 8). Functional integrity of the single-substrate sensor was further validated by the addition of the self-inserting nanopore α-hemolysin at saturating peptide concentrations, which triggered a complete transition to the high-FRET state (Supplementary Fig. 6e), demonstrating full sensor functionality inside liposomes.

To validate peptide transport independently of fluorescence, we established a label-free workflow using liquid chromatography-coupled mass spectrometry (LC-MS). LC-MS confirmed ATP-dependent uptake of unlabeled peptides used in single-molecule assays (Supplementary Fig. 9).

### Antigenic peptide transport by single transport complexes

Liposomes containing a single TmrAB$^{WT}$ were immobilized via a nanobody that captures the transport complex in an uptake-competent orientation. In the absence of Mg-ATP and peptide, the single-molecule sensor OppA encapsulated inside liposomes remained in a low-FRET state ($E = 0.6$) for ≥80 min at 30 °C (Fig. 2a). Likewise, in the presence of Mg-ADP (3 mM) and peptide (200 μM), no transport occurred over 50 min (Fig. 2b), confirming that Mg-ADP does not support transport and that liposomes are not peptide-leaky. Upon addition of 3 mM Mg-ATP and 200 μM peptide at 30 °C, the sensor gradually shifted to the high-FRET state over several minutes (Fig. 2c), reporting peptide translocation by individual wild-type transporters.

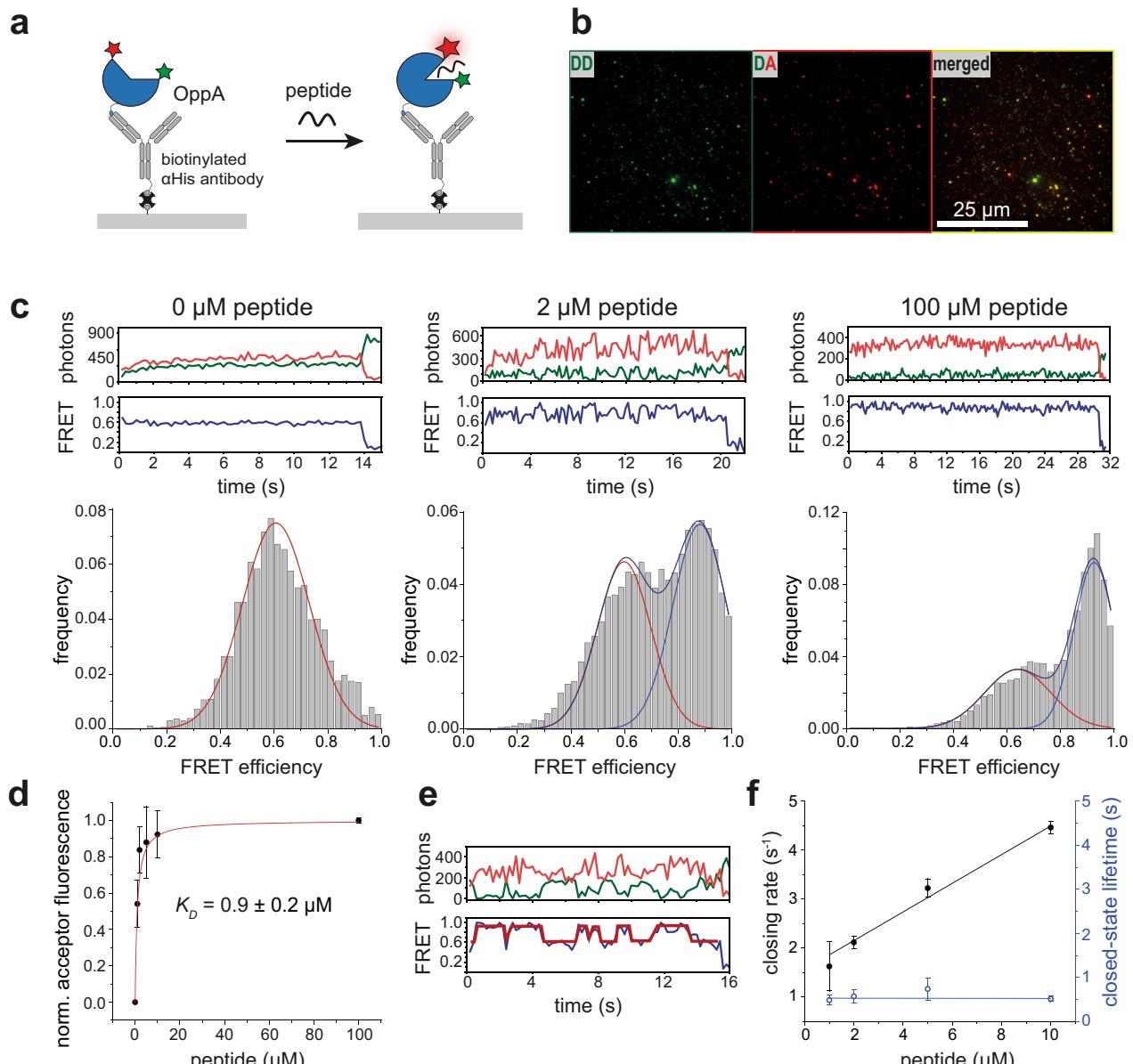

**Fig. 1 | Single-molecule FRET sensor for antigenic peptide detection.**
**a** Schematic of surface-immobilized OppA illustrating open (apo) and closed (holo) conformations upon binding of the peptide RRYQKSTEL. High FRET efficiency is depicted by a brighter acceptor fluorophore (red star) relative to the donor (green star). **b** Fluorescence micrographs of immobilized peptide sensors showing donor emission upon donor excitation (DD), acceptor emission upon donor excitation (DA), and merged colocalization of donor and acceptor signals. **c** Representative donor (green) and acceptor (red) fluorescence traces with corresponding FRET trace (blue). Histograms below display FRET efficiency distributions of static and dynamic smFRET traces at increasing peptide concentrations, showing low-FRET ($E = 0.6$, red), high-FRET ($E = 0.9$, blue), and mixed populations (blue). Left: $n = 317$

molecules, middle: $n = 242$, right: $n = 132$. **d** Acceptor fluorescence under donor excitation, quantified as the area of the low-FRET Gaussian (red) relative to the high-FRET Gaussian (blue) across peptide concentrations normalized to the highest peptide concentration. Fitting yields an equilibrium dissociation constant $K_D$ of $0.9 \pm 0.2\,\mu M$. Data represent mean ± s.d. from $n = 3$ independent experiments. **e** Representative dynamic single-molecule FRET trace with step-function fitting used to extract transition kinetics. **f** Peptide concentration-dependent closing rates (black; $k_{on} = 0.29 \pm 0.02\,\mu M^{-1}s^{-1}$) and closed-state lifetime (blue; $\tau = 0.5 \pm 0.1\,s$), extracted from $n = 1311$ single-molecule traces. Data represent mean ± s.d. from $n = 3$ independent experiments.

## Single-substrate translocation by individual ABC transporters

We next used the slow-turnover variant TmrA$^{EQ}$B, which exhibits a prolonged OF lifetime of ~25 min at 20 °C[33], enabling temporal resolution of individual transport events. Neither Mg-ADP nor nucleotide-free conditions resulted in peptide transport (Fig. 3a, b). At 45 °C for 5 min, peptide translocation was observed only in the presence of Mg-ATP (Fig. 3c). After a first transport event, removal of free peptides and nucleotides and imaging at 30 °C allowed the transporter to relay back to the IF state[33]. A second Mg-ATP/peptide incubation induced a second transport event, evident from an increased high-FRET population

(Fig. 3d, e). Control experiments at 30 °C and in the presence of valinomycin excluded effects from a transmembrane potential (Supplementary Fig. 10a–e). A control experiment without Mg-ATP after the first transport event confirmed the absence of liposome leakiness (Supplementary Fig. 10f).

The liposome size distribution immobilized on the microscope surface was assessed by staining with the membrane-associated dye DiD and super-resolution imaging via direct stochastic optical reconstruction microscopy (dSTORM), revealing that most liposomes had a diameter of ~100 nm (Fig. 3g, Supplementary Fig. 11). HMM analysis of

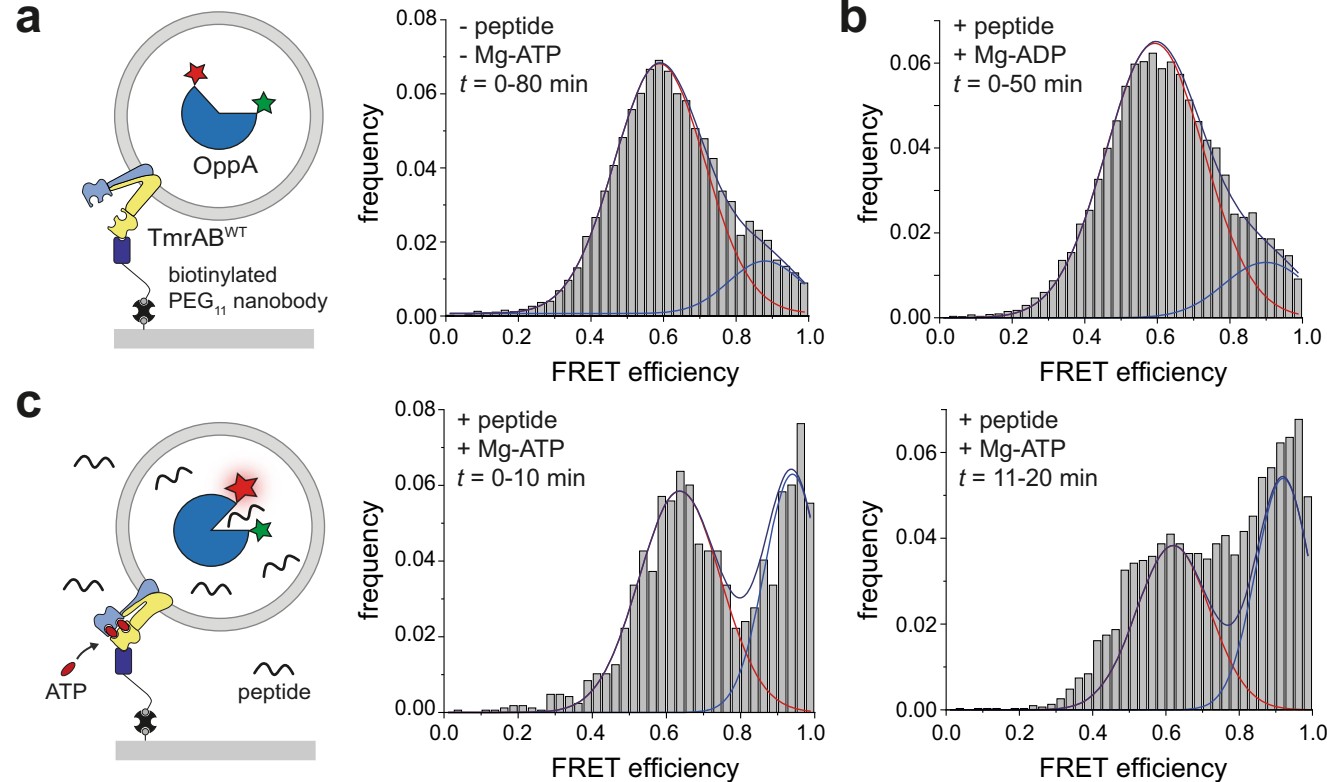

**Fig. 2 | Peptide translocation by a single ABC transporter in an uptake-competent orientation. a** Schematic of a single wild-type TmrAB (TmrAB[WT]) proteoliposome immobilized on a streptavidin-functionalized glass slide via the conformationally non-selective nanobody Nb9F10 linked to a biotinylated PEG[11] linker (left). OppA and TmrAB are not drawn to scale. Right: FRET efficiency distribution of encapsulated OppA in the absence of peptide and Mg-ATP, showing the apo state ($E = 0.6$; $n = 184$ molecules). **b** FRET distribution after addition of Mg-ADP (3 mM) and peptide (200 μM RRYQKSTEL), showing no detectable peptide transport ($n = 214$). **c** Schematic of peptide transport into the liposome lumen (left). FRET distributions of OppA following addition of Mg-ATP (3 mM) and peptide (200 μM RRYQKSTEL). Peptide arrival in the lumen is indicated by a shift in OppA from the low-FRET state ($E = 0.6$) to the high-FRET state ($E = 0.9$) over time. Middle histogram: 0–10 min, $n = 49$; right histogram: 11–20 min, $n = 66$. Histograms represent data from $n = 3$ independent experiments.

smFRET dynamics allowed us to calculate the OppA closing rate and quantify lumenal peptide concentrations of $4 \pm 1$ μM and $6 \pm 1$ μM after the first and second translocation events, respectively (Figs. 1f and 3f). These measured lumenal peptide concentrations closely match the theoretical expectations for one and two peptides per 100 nm liposomes (Fig. 3g, Supplementary Fig. 11). For the smFRET measurements, larger liposomes containing multiple FRET sensors were excluded from our analysis. In summary, these data demonstrate that ATP binding alone is sufficient to drive a single peptide-transport event via TmrA[EQ]B.

## ATP binding alone without Mg²⁺ triggers the outward-facing conformation

Adding Mg-ATP to TmrAB induces the transition from the IF to the OF conformation, as shown by previous functional and structural studies[14,33]. To test whether ATP alone without Mg²⁺ can drive the conformational switch, wild-type TmrAB[WT] and the slow-turnover variant TmrA[EQ]B were reconstituted into lipid nanodiscs, incubated with ATP-EDTA and the peptide, and analyzed by single-particle cryo-EM (Supplementary Figs. 12 and 13). At 3.02 Å resolution, both TmrAB[WT] and TmrA[EQ]B adopt an outward-facing occluded (OF[occluded]) conformation (Fig. 4a–c, Supplementary Fig. 14 and Supplementary Table 1). Both structures closely resemble the Mg-ATP-bound OF[occluded] conformation of TmrA[EQ]B (PDB 6RAI; EMD-4776)[14], with root mean square deviations across all Cα atoms of 0.51 Å for TmrA[EQ]B and 0.53 Å for TmrAB[WT]. Two ATP molecules are bound at the canonical and non-canonical nucleotide-binding sites (NBSs), respectively (Fig. 4b, c, Supplementary Fig. 14). Unlike in the Mg-ATP-bound structure[14],

however, the ATP-EDTA complexes lack Mg²⁺ coordination: key residues, Q419 and S378 in TmrB, and Q441 and T400 in TmrA, no longer position Mg²⁺ to coordinate the β- and γ-phosphates of ATP (Fig. 4b, c). A difference EM map comparing TmrA[EQ]B structures in the OF[occluded] state with Mg-ATP or ATP-EDTA (EMD-4776 vs. EMD-54377) confirms the absence of Mg²⁺ at both NBSs (Fig. 4d). Notably, no density corresponding to a bound peptide was observed in the cryo-EM map of the OF[occluded] conformation.

Finally, radiometric ATPase assays performed at 45 °C showed that ATP hydrolysis by wild-type TmrAB (Fig. 4e) and the slow-turnover variant (Fig. 4f) strictly depends on Mg²⁺. In the presence of 5 mM MgCl₂, robust ATP hydrolysis of ∼1 s⁻¹ for TmrAB[WT] and ∼0.25 min⁻¹ for TmrA[EQ]B was observed, whereas Mg²⁺ chelation with 10 mM EDTA abolished the ATPase activity (Fig. 4e, f; Supplementary Fig. 15).

## ATP binding without Mg²⁺ drives single-substrate translocation

To clarify the role of Mg²⁺, we first performed ensemble assays. TmrAB[WT] reconstituted into liposomes showed no detectable transport in the presence of ATP-EDTA, as assessed by both fluorescence-based and LC-MS readouts (Supplementary Fig. 16a, b). In contrast, at the single-molecule level, however, liposomes containing a single reconstituted TmrAB[WT] transporter and pre-incubated with EDTA (10 mM) for 5 min exhibited a single-transport event upon addition of ATP-EDTA and peptide (Supplementary Fig. 16c, e). No additional rounds of transport were detected under these conditions, and multiple turnovers strictly required the presence of Mg²⁺ (Supplementary Fig. 16d).

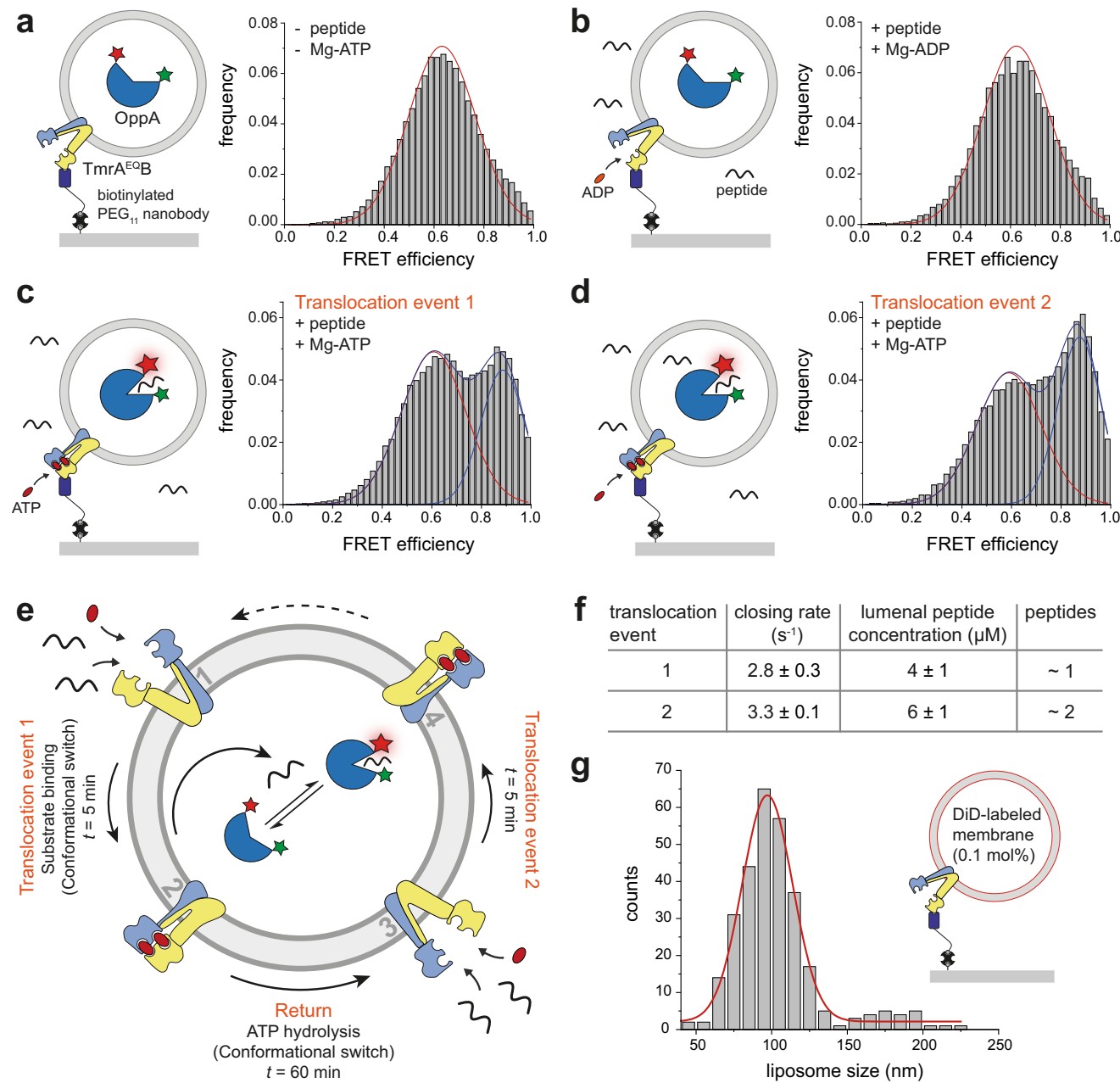

**Fig. 3 | Single-molecule resolution of sequential peptide transport by the slow-turnover TmrA^(EQ)B variant. a** Schematic of OppA in the apo (unbound) state (left). In the absence of Mg-ATP and peptide, OppA predominantly adopts a low-FRET state ($E = 0.6$), as shown in the corresponding histogram ($n = 898$ molecules, right). **b** After addition of Mg-ADP (3 mM) and peptide (200 μM RRYQKSTEL), no transport occurs, and OppA remains in the low-FRET state ($E = 0.6$; right). Excess peptide and nucleotides were removed before data acquisition ($n = 170$ molecules). **c** Schematic of the first peptide translocation event (left). A single-transport event is triggered by a 5-min incubation with Mg-ATP (3 mM) and peptide (200 μM RRYQKSTEL) at 45 °C, followed by removal of unbound ligands. FRET traces recorded over 60 min at 30 °C show a population shift to the high-FRET state ($E = 0.9$), consistent with successful peptide translocation into the liposome lumen ($n = 820$). **d** A second peptide translocation event is initiated by a second exposure to Mg-ATP and peptide, yielding an additional increase in the high-FRET population ($n = 689$). Histograms represent pooled data from $n \geq 3$ independent

experiments. **e** Model of the TmrA^(EQ)B transport cycle during two successive peptide transport events. Upon addition of peptide and Mg-ATP for 5 min, the transporter switches from the inward-facing state (1) to the outward-facing state (2), resulting in single-substrate translocation. During subsequent imaging, ATP hydrolysis returns the transporter to the inward-facing state (3). A second 5-min incubation with peptide and Mg-ATP initiates a second transport event (3→4). **f** Hidden Markov Model (HMM) analysis of smFRET trajectories reveals distinct OppA peptide-binding kinetics, with closing rates of $2.8 \pm 0.3\ \mathrm{s^{-1}}$ for the first translocation event ($n = 395$) and $3.3 \pm 0.1\ \mathrm{s^{-1}}$ for the second ($n = 629$). These rates correspond to apparent lumenal peptide concentrations of $4 \pm 1\ \mathrm{\mu M}$ and $6 \pm 1\ \mathrm{\mu M}$, respectively. **g** Liposomes containing individual TmrAB were captured via the conformationally non-selective nanobody Nb9F10 in an uptake-competent orientation. The lipid membrane was labeled by the lipophilic dye DiD (0.1 mol%) and imaged by *d*STORM (Supplementary Fig. 11). Histogram analysis of liposome diameters revealed that the predominant population has a diameter of ~100 nm.

Similarly, TmrA^(EQ)B liposomes pre-incubated with 10 mM EDTA for 5 min showed no translocation, either in the absence of nucleotide and peptide or when exposed to peptide with ADP-EDTA (Supplementary Fig. 16f, g). Addition of peptide together with ATP-EDTA initiated a

transition of OppA to the high-FRET state, indicating successful translocation of a single peptide (Fig. 5a). A second ATP-EDTA addition did not elicit further transport (Fig. 5b), consistent with cryo-EM data showing the transporter locked in the OF state. After an ATP-EDTA-

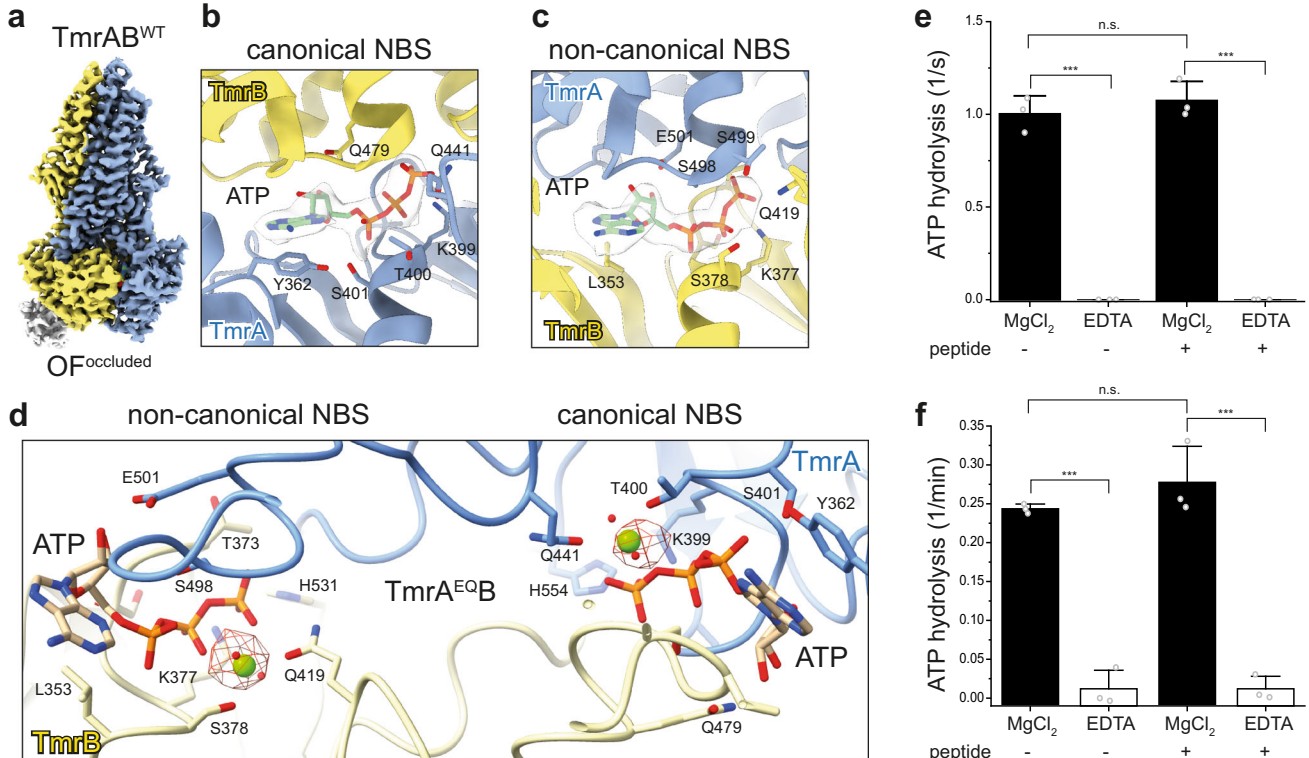

**Fig. 4 | ATP-EDTA triggers NBD dimerization and conformational switching in TmrAB. a** Cryo-EM structure of wild-type TmrAB (TmrAB[WT]) in the presence of peptide and ATP-EDTA reveals a transition to the outward-facing occluded (OF[occluded]) conformation, resolved at 3.02 Å (EMD-54378, PDB 9RYF). Cryo-EM maps show ATP bound at both the canonical (**b**) and non-canonical (**c**) nucleotide-binding sites (NBSs), despite the absence of Mg[2+]. **d** Structural comparison of the OF[occluded] conformations of TmrA[EQ]B bound to Mg-ATP (EMD-4776, PDB 6RAI)[14] versus ATP-EDTA resolved at 3.02 Å (EMD-54377, PDB 9RYE). The difference map (red mesh) highlights the loss of Mg[2+] density at both NBSs. Key ATP-interacting residues are shown as sticks and are labeled. ATPase activity of (**e**) TmrAB[WT] and (**f**) TmrA[EQ]B demonstrates that ATP hydrolysis is strictly Mg[2+]-dependent. Proteoliposomes containing TmrAB were incubated with 0.3 mM [γ[32]P]-ATP for 18 min at 45 °C in the presence of 5 mM MgCl$_2$ or 10 mM EDTA. ATP autohydrolysis controls lacked a transporter. Release of [γ−[32]P] was quantified by thin-layer chromatography and autoradiography. Statistical significance was assessed by two-way ANOVA ($n = 3$ independent experiments). ***$P \leq 0.0001$; n.s. not significant. Bars represent means ± s.d.

driven translocation event and a 60-min relaxation period, addition of Mg-ATP and peptide restored transporter activity (Fig. 5c, d), confirming that Mg[2+] is required to reset the transporter.

Together, these findings demonstrate that ATP binding alone is sufficient to drive a single-substrate translocation event in both TmrAB[WT] and TmrA[EQ]B, while sustained transport cycles strictly require Mg[2+]-dependent ATP hydrolysis.

## Discussion

To directly observe substrate translocation by individual ABC transporters, we developed a single-molecule platform that combines a FRET-based peptide sensor encapsulated inside liposomes with transporter immobilization in an uptake-competent orientation using a conformationally non-selective nanobody. This approach enables real-time monitoring of the activity of uptake-competent transporters at the single-molecule level. Using a slow-turnover variant of TmrAB (TmrA[EQ]B), we detected discrete, quantized transport events arising from individual transport complexes.

A central and surprising finding is that ATP binding alone, even in the absence of Mg[2+], can drive substrate translocation by triggering a conformational switch from the IF to the OF state in TmrAB. In TmrA[EQ]B, each ATP binding event was tightly coupled to a single peptide transport step—a level of resolution that goes far beyond what is possible with conventional bulk assays. Traditional liposome-based transport assays rely on filtration or centrifugation followed by quantification of substrate uptake via radioactivity or fluorescence[22,34–36]. Such approaches suffer from two major limitations: First, chemical

labeling or modification of peptides can perturb binding affinity (sometimes increasing them)[13]. Second, ensemble averaging obscures functional heterogeneity, in particular variation in transporter orientation and activity[37,38]. To overcome these limitations, alternative single-liposome techniques such as dual-color fluorescence burst analysis and flow cytometry-based assays have been developed[33,39,40]. While these techniques offer insights into liposome-level heterogeneity and coupling stoichiometries, they still rely on labeled substrates and lack single-translocation resolution.

Our single-molecule approach allowed us to dissect the specific role of Mg[2+] in the transport cycle. We found that while Mg[2+] is not strictly required for ATP binding or the IF-to-OF conformational transition (as shown in ATP-EDTA-driven translocation), it is essential for ATP hydrolysis and for resetting the transporter to its resting state. These observations are consistent with earlier reports that ATP hydrolysis triggers the return to the IF conformation via "unlocked return" intermediates[14]. Based on our data, we propose the following mechanistic model (Fig. 6): (i) ATP binding induces NBD dimerization and the IF-to-OF conformational switch; (ii) peptide translocation occurs concurrently; and (iii) Mg[2+]-dependent ATP hydrolysis resets the transporter for another cycle.

Our results extend previous work on ABC transporters and add mechanistic insights specific to heterodimeric type IV ABC systems. Computational studies have suggested that Mg[2+] primarily accelerates ATP hydrolysis rather than being strictly required for ATP binding[41,42]. For instance, in the Cystic Fibrosis Transmembrane Conductance Regulator (CFTR), ATP binds to the non-canonical site independently

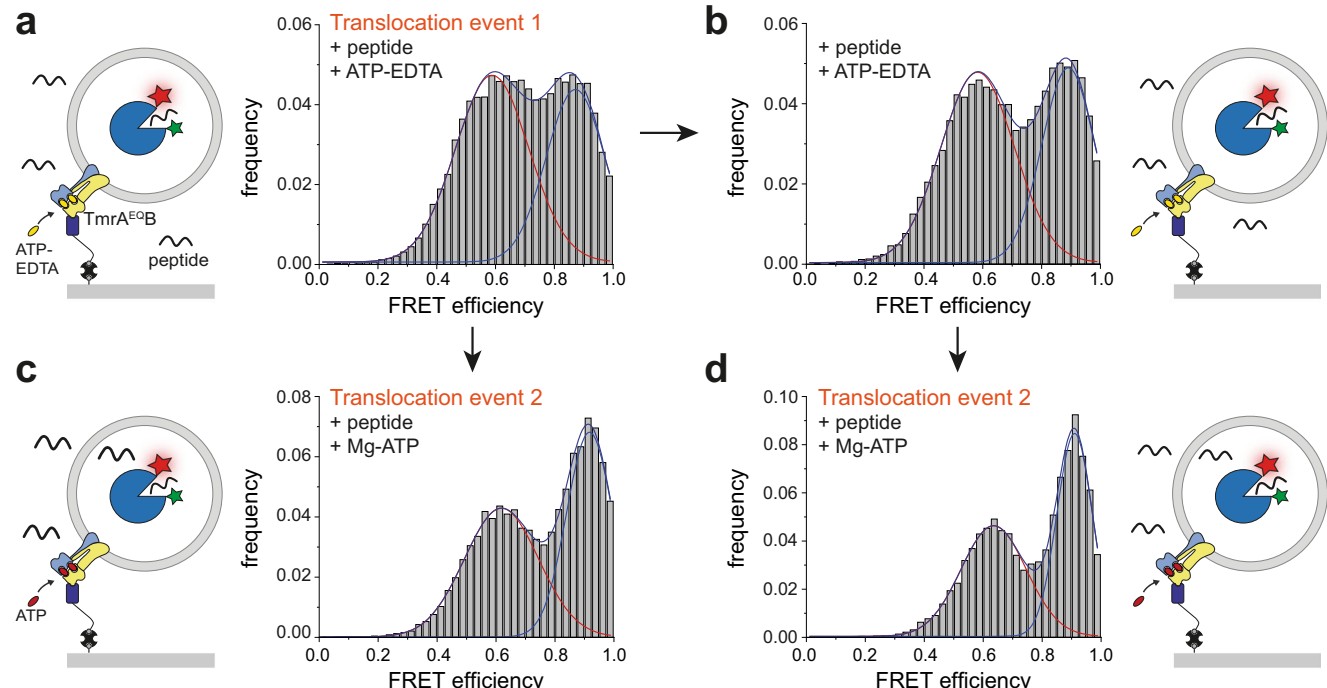

**Fig. 5 | ATP is sufficient to trigger a single-peptide translocation event in the absence of Mg²⁺. a** First translocation event using the slow-turnover variant TmrA$^{EQ}$B. Proteoliposomes were incubated with ATP-EDTA (3 mM) and peptide (200 μM RRYQKSTEL) for 5 min at 45 °C. Excess nucleotide and peptide were removed prior to data acquisition, which was performed over 60 min at 30 °C (*n* = 329 molecules). **b** A second incubation with fresh ATP-EDTA (3 mM) and peptide (200 μM RRYQKSTEL) under identical conditions did not increase the high-FRET population, indicating that additional transport events do not occur in the absence of Mg²⁺ (*n* = 323). **c** Left: Schematic of the second translocation event initiated by Mg-ATP. Right: After the initial ATP-EDTA incubation (**a**), addition of Mg-ATP (3 mM) and peptide (200 μM RRYQKSTEL) enabled a second peptide translocation event (*n* = 270). **d** After two successive ATP-EDTA incubations, subsequent addition of Mg-ATP and peptide restored transport activity, demonstrating that the transporter regains full transport competence (*n* = 238). The schematic summarizes the complete experimental sequence. Histograms represent data from *n* = 3 independent experiments.

of Mg²⁺, whereas ATP binding and hydrolysis at the canonical site remain Mg²⁺-dependent[19]. These observations collectively indicate that Mg²⁺ dependence may vary between nucleotide-binding sites, particularly in heterodimeric ABC transporters. Consistently, a previous study of a D-loop mutant in human TAP1 showed that ATP hydrolysis is not required for substrate translocation[22].

TmrAB is a heterodimeric type IV ABC transporter containing one canonical and one non-canonical NBS[3]. In such heterodimers, only the canonical NBS is catalytically competent and hydrolyzes ATP, whereas the non-canonical NBS is degenerated. Current models propose that ATP binding at both sites drives the IF-to-OF transition, while ATP hydrolysis at the canonical site resets the transporter[14,43]. Notably, to our knowledge, an ATP-EDTA structure or ATP-EDTA-dependent transport activity has not previously been reported or analyzed at the single-molecule level. Heterodimeric ABC transporters typically exhibit strong allosteric coupling between their non-canonical and canonical NBSs. In addition, the ATP analog AMP-PNP preferentially binds to the non-canonical site, which likely explains the incomplete NBD dimerization and the lack of a conformational switch in the TMDs[44–46]. These structural observations are consistent with our smFRET data (Supplementary Fig. 17).

We acknowledge several limitations in our study. First, we were unable to perform single-molecule measurements on wild-type TmrAB under fully physiological conditions, because its optimal activity occurs above 45 °C. These temperatures exceed both the technical limits of our microscope and the thermal stability range of the OppA-based smFRET sensor, which begins to unfold at >45 °C[47]. Second, this constraint required us to conduct transport measurements at two different temperatures—30 °C for TmrAB$^{WT}$ and 45 °C for TmrA$^{EQ}$B— preventing a direct quantitative comparison of their transport kinetics. Third, the OppA smFRET sensor saturates after a few uptake events,

which limits our ability to determine whether additional transporter cycles occur in discrete bursts or proceed continuously. This phenomenon is reminiscent of a "resting" state reported for other molecular machines, such as the rotary V-ATPase[26]. Finally, while liposome size control after reconstitution is inherently limited, the applied protocols reliably generate predominantly unilamellar vesicles[48]. Importantly, the assay design—using very low transporter-to-lipid ratios and orientation-specific surface immobilization—selectively removes empty, oversized, and non-functional liposomes, resulting in a highly enriched population of functional proteoliposomes with an apparent size of ~100 nm (Supplementary Fig. 11).

Another unresolved question concerns the coupling stoichiometry between ATP hydrolysis and substrate translocation. While the slow-turnover variant TmrA$^{EQ}$B shows a tight 1:1 coupling, the wild-type transporter exhibits significant futile ATPase activity[13]. Intriguingly, ATP-EDTA appears to enforce single-turnover behavior even in the wild-type transporter; however, the broader principles that govern coupling stoichiometry across different ABC transporters remain poorly understood[1,49–52].

In conclusion, our work establishes a robust single-molecule mechanistic framework for understanding substrate transport in TmrAB, a heterodimeric ABC transporter. Beyond offering deeper mechanistic insight, this approach also opens the door to broader applications: we envision using similar platforms to dissect coupling, kinetics, and conformational dynamics in other transport systems— one molecule at a time.

## Methods

### Production and purification of *Lactococcus lactis* OppA

The double-cysteine variant *Lactococcus lactis* OppA$^{A209C/S441C}$ (*Ll*OppA$^{AS}$)[30] was expressed in the *Escherichia coli* MC1061 grown in

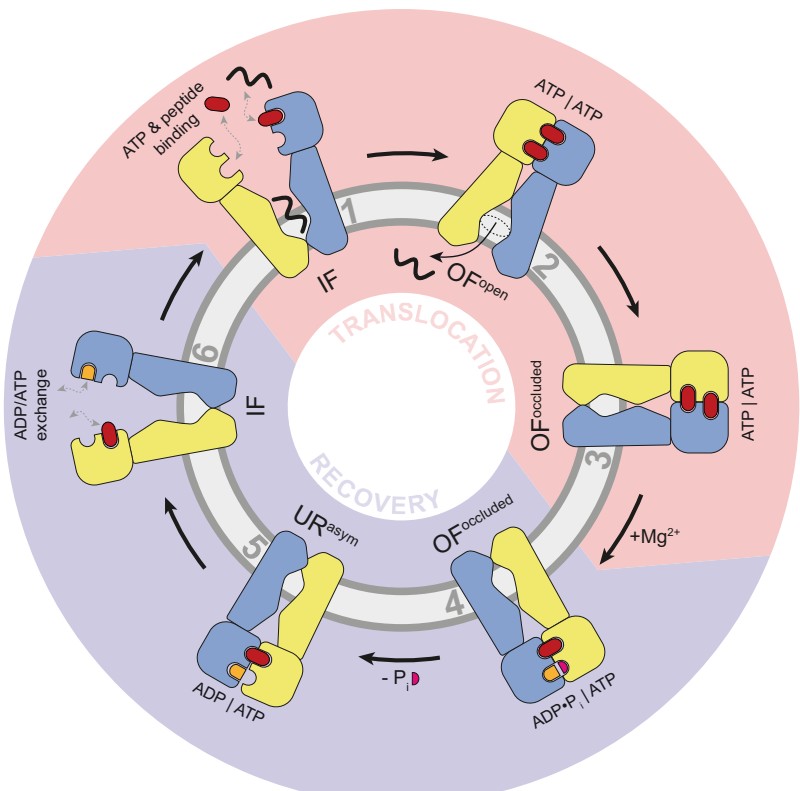

**Fig. 6 | Model of peptide translocation by the heterodimeric ABC transporter TmrAB.** In the inward-facing (IF) conformation (**1**), the transporter binds peptide substrate and exchanges nucleotides (ADP and ATP). ATP binding promotes nucleotide-binding domain (NBD) dimerization, driving the transition to the outward-facing (OF) state (**2–3**) and enabling peptide translocation across the membrane. This ATP-driven translocation step is indicated by the light red shading. $Mg^{2+}$ is required for ATP hydrolysis in the OF state (**3**), resulting in phosphate release (**4**) and resetting the transporter to the IF state (**5**). Completion of these steps constitutes the full transport cycle (**6**), highlighted in light purple.

Terrific Broth (TB) supplemented with 100 µg/mL ampicillin. Cultures were incubated at 37 °C to an $OD_{600}$ of 0.4, cooled to 25 °C, and induced at an $OD_{600}$ of 0.6–0.8 with 0.1% (w/v) L-arabinose. Following overnight expression, cells were harvested (4500 × $g$, 4 °C, 15 min) and stored at −80 °C. All subsequent purification steps were performed at 4 °C. Cell pellets were resuspended in purification buffer (25 mM potassium phosphate ($KP_i$), pH 6.5, 100 mM KCl, 15 mM imidazole, 0.5 mM PMSF, 1 mM DTT) and lysed by sonication (Branson Sonifier 250). The lysate was clarified (18,000 × $g$, 35 min, 4 °C) and applied to Ni-NTA affinity chromatography (Bio-Rad) pre-equilibrated in equilibration buffer (25 mM $KP_i$, pH 6.5, 100 mM KCl, 15 mM imidazole, 1 mM DTT). After washing with 10 column volumes (CV) of equilibration buffer and 10 CV of wash buffer (25 mM $KP_i$ pH 6.5, 100 mM KCl, 30 mM imidazole, 1 mM DTT), the protein was eluted with 8 CV of elution buffer (25 mM $KP_i$ pH 6.0, 20 mM KCl, 400 mM imidazole, 1 mM DTT). Further purification was performed by cation exchange chromatography (HiTrap SP HP, Cytiva) with a linear gradient from buffer A (25 mM $KP_i$ pH 6.0, 20 mM KCl, 1 mM DTT) to buffer B (25 mM $KP_i$ pH 6.0, 500 mM KCl, 1 mM DTT). To remove residual endogenous ligand, *Ll*OppA$^{AS}$ underwent partial unfolding and refolding by stepwise dialysis in decreasing concentrations of guanidine hydrochloride (GuHCl): 2 M (1 h, 8 °C), 1.5 M, 1.0 M, 0.5 M each 30 min, 8 °C. The purified protein was labeled and stored at 4 °C.

### Fluorophore labeling of *Ll*OppA$^{AS}$
Site-specific labeling was performed via maleimide chemistry using Alexa Fluor 555 (AF555) and Alexa Fluor 647 (AF647; Thermo Fisher Scientific). Protein and dyes were mixed at a 1:10:10 molar ratio in OppA-SEC buffer (25 mM HEPES-NaOH, pH 7.5, 300 mM NaCl, and 10% (v/v) glycerol) at 8 °C for 18 h. Excess dyes were quenched with 2 mM β-

mercaptoethanol and removed by Zeba™ spin desalting (Thermo Fisher Scientific), followed by size-exclusion chromatography (SEC, Superdex 200 Increase 3.2/300 column, Cytiva) in OppA-SEC buffer. Labeling efficiency was determined from the SEC absorbance at 280 nm, 550 nm, and 650 nm.

### Ensemble fluorescence analysis of OppA
Peptide binding of unlabeled or AF555/AF647-labeled OppA was monitored using a Varian Cary Eclipse scanning fluorimeter. Excitation/emission slit widths were set to 5/10 nm. All measurements were performed at room temperature in buffer containing 25 mM HEPES-NaOH, pH 7.5, 150 mM NaCl, 5% (v/v) glycerol. For ensemble FRET experiments, OppA (150 nM) was excited at $\lambda_{ex}$ = 530 nm, and fluorescence emission spectra were recorded. Tryptophan quenching experiments were performed to probe peptide binding to unlabeled *Ll*OppA$^{AS}$. Samples were excited at 280 nm, and emission spectra were recorded during titration with increasing peptide concentrations (0.1–10 µM). For titration, OppA was preincubated with either 5 µM or 100 µM RRYQKSTEL.

### Production and purification of TmrAB
Recombinant expression and purification of His$_{10}$-tagged wild-type TmrAB (TmrAB$^{WT}$) and slow-turnover variant TmrA$^{E523Q}$B (TmrA$^{EQ}$B) were performed as described previously[9,13,14,33,53]. *E. coli* BL21 (DE3) cells were cultured in LB high-salt medium supplemented with 100 µg/mL ampicillin at 37 °C. Upon reaching an $OD_{600}$ of 0.6, protein expression was induced with 0.5 mM isopropyl-β-D−1-thiogalactopyranoside (IPTG), followed by incubation for 18 h at 20 °C. Cells were harvested by centrifugation (4500 × $g$, 4 °C, 15 min) and resuspended in lysis buffer (20 mM HEPES-NaOH, pH 7.5, 300 mM NaCl, 0.2 mM PMSF).

Cells were lysed by sonication, and debris was removed by centrifugation (18,000 × $g$, 35 min, 4 °C). Membranes were pelleted (100,000 × $g$, 2 h, 4 °C), resuspended in lysis buffer, and solubilized for 1 h at 4 °C with 20 mM β-$n$-dodecyl β-D-maltoside (β-DDM). The solubilized fraction was cleared by ultracentrifugation (100,000 × $g$, 30 min, 4 °C), and the supernatant was incubated with Ni-NTA agarose resin (Qiagen) for 1 h at 8 °C. After washing with 10 CV of wash buffer (20 mM HEPES-NaOH pH 7.5, 300 mM NaCl, 1 mM β-DDM, 50 mM imidazole), the protein was eluted with elution buffer (20 mM HEPES-NaOH pH 7.5, 300 mM NaCl, 1 mM β-DDM, 300 mM imidazole). Eluted fractions were buffer-exchanged into TmrAB-SEC buffer (20 mM HEPES-NaOH, pH 7.5, 150 mM NaCl, 1 mM β-DDM) using a PD-10 desalting column (Cytiva), followed by size-exclusion chromatography (SEC) on a Superdex 200 Increase 10/300 GL column (Cytiva) in the same buffer. Fractions containing monodisperse TmrAB were pooled, flash-frozen in liquid nitrogen, and stored at −80 °C in TmrAB-SEC buffer supplemented with 20% (v/v) glycerol.

## Nanobody production and purification

The nanobody Nb9F10$^{S63C}$ was expressed in $E.~coli$ BL21 (DE3) in TB supplemented with 100 μg/mL ampicillin at 37 °C, as described previously[14]. Expression was induced by the addition of 1 mM IPTG at an $OD_{600}$ of 0.6, and cells were grown overnight at 28 °C. Cells were harvested by centrifugation (4500 × $g$, 4 °C, 15 min) and pellets were stored at −80 °C. For purification, pellets were resuspended in nanobody lysis buffer (25 mM HEPES-NaOH, pH 7.4, 300 mM NaCl, 15 mM imidazole, 0.5 mM PMSF) and lysed by sonication. The lysate was clarified by centrifugation (18,000 × $g$, 35 min, 4 °C) and applied to Ni-NTA resin (Bio-Rad) equilibrated in 25 mM potassium phosphate ($KP_i$) pH 6.5, 100 mM KCl, 20 mM imidazole, and 0.5 mM TCEP. After washing with 10 CV of the same buffer, the bound nanobody was eluted with 8 CV of 25 mM $KP_i$, pH 6.0, 20 mM KCl, 300 mM imidazole, and 0.5 mM TCEP. Eluted fractions were pooled and further purified by cation exchange chromatography using a HiTrap SP HP column (Cytiva) with a linear gradient from low (25 mM $KP_i$ pH 6.0, 20 mM KCl, 0.5 mM TCEP) to high salt buffer (25 mM $KP_i$ pH 6.0, 500 mM KCl, 0.5 mM TCEP). The nanobody was concentrated and subjected to SEC on a Superdex 200 Increase 10/300 GL column (Cytiva). Site-specific coupling of the nanobody Nb9F10 to biotin-PEG$_{11}$-maleimide (Sigma-Aldrich) or AF555 (Thermo Fisher Scientific) was performed in the presence of 0.5 mM TCEP at 8 °C for 2 h, using nanobody-to-compound molar ratios of 1:1.2 and 1:5, respectively. Excess linker or fluorophore was removed by Zeba™ spin desalting (Thermo Fisher Scientific), followed by SEC on a Superdex 200 Increase 10/300 GL column (Cytiva).

## SDS-PAGE, in-gel fluorescence, and immunoblotting

Protein purity and fluorescence labeling were analyzed by SDS-PAGE. Separation gels (12% (w/v) acrylamide) were prepared with 1.5 M Tris-HCl, pH 8.8, 0.4% (w/v) SDS, 0.05% (w/v) ammonium persulfate (APS), and 0.25% (v/v) $N,N,N',N'$-tetramethylethylenediamine (TEMED). Stacking gels (5% (w/v) acrylamide) were prepared with 0.5 M Tris-HCl, pH 6.8, 0.4% (w/v) SDS, 0.1% (w/v) APS, and 0.03% (v/v) TEMED. Gels were used immediately or stored at 4 °C for up to 4 weeks. Protein samples were mixed with 4x Laemmli loading buffer containing dithiothreitol (DTT) and heated to 95 °C for 10 min. Electrophoresis was performed in SDS running buffer (25 mM Tris-HCl, pH 8.3, 192 mM glycine, 0.1% (w/v) SDS) at 180 V. For protein visualization, gels were stained with InstantBlue™ Protein Stain (Expedeon) for 1 h at room temperature with gentle agitation and imaged using the Fusion FX imaging system (Vilber). In-gel fluorescence was recorded using excitation at 555 nm for AF555 (filter set F-595 Y3) and 640 nm for AF647 (filter set F-710). Exposure times were optimized individually for each fluorophore.

Proteins separated by SDS-PAGE were transferred to polyvinylidene fluoride (PVDF) membranes by wet electroblotting in phosphate-buffered saline (PBS) pH 8.4 supplemented with 0.1% (w/v) SDS and 20% (v/v) methanol at 30 V for 15 h at 4 °C. Membranes were blocked in PBS supplemented with 0.05% (v/v) Tween-20 (PBS-T) and 5% (w/v) non-fat dry milk for 1 h at room temperature, then incubated with horseradish peroxidase (HRP)-conjugated anti-6xHis (anti-6×His tag® antibody, Abcam, 1:2000 dilution) in blocking solution for 1 h at 4 °C. After three washes with PBS-T, membranes were developed using enhanced chemiluminescence (Clarity Western ECL Substrate, Bio-Rad) and imaged on a Fusion FX imaging system (Vilber).

## Liposome preparation

Liposomes were prepared by mixing $E.~coli$ polar lipids and 1,2-dioleoyl-sn-glycero-3-phosphocholine (DOPC) (both from Avanti Polar Lipids) at a molar ratio of 7:3 in chloroform to a final concentration of 5 mg/mL. The lipid mixture was dried under vacuum using a rotary evaporator and subsequently resuspended in liposome buffer (25 mM HEPES-NaOH, pH 7.5, 150 mM NaCl, and 5% (v/v) glycerol). Large unilamellar vesicles (LUVs) were generated by sonication for 30 min, followed by five freeze-thaw cycles. Liposomes were stored at −80 °C. Before use, LUVs were extruded 11 times through 100 nm polycarbonate membranes using a LiposoFast-Basic extruder (Avestin).

## Reconstitution of TmrAB and encapsulation of the single-molecule sensor

TmrAB$^{WT}$ and the slow-turnover variant TmrA$^{E523Q}$B (TmrA$^{EQ}$B) were reconstituted into liposomes composed of $E.~coli$ polar lipids and DOPC at a molar ratio of 7:3. For single-molecule experiments, a protein-to-lipid ratio (w/w) of 1:10,000 and an OppA-to-liposome ratio of 0.2 molecules per liposome were used. For ensemble transport assays, a protein-to-lipid ratio (w/w) of 1:20 was applied. Extruded LUVs were destabilized by incubation with 0.3% (v/v) Triton X-100 (TX-100) for 30 min. TmrAB and OppA were then added to the detergent-destabilized liposomes and incubated for 30 min at 8 °C under gentle rotation. Detergent removal was performed gradually using sequential additions of polystyrene Bio-Beads SM-2 (Bio-Rad): two incubations at 40 mg/mL (1 h and overnight), followed by two incubations at 80 mg/mL (1 h each). Finally, proteoliposomes were harvested by ultracentrifugation (100,000 × $g$, 30 min, 4 °C) and resuspended in the appropriate buffer.

## Quantification of stochastic TmrAB reconstitution

TmrAB$^{WT}$ was labeled with AF647 via maleimide chemistry using an intrinsically occurring single-cysteine residue. Labeling was performed at a 1:10 molar ratio of protein to dye for 2 h at 8 °C on an overhead rotor in TmrAB-SEC buffer. Unreacted fluorophore was removed by SEC in TmrAB-SEC buffer. The labeled TmrAB was reconstituted into liposomes at a protein-to-lipid ratio of 1:10,000 (w/w) as described above. Liposomes containing AF647-labeled TmrAB$^{WT}$ were immobilized using a biotinylated PEG$_{11}$-conjugated nanobody. Data were acquired using a TIRF microscope (NanoImager S, Oxford Nanoimaging). For each region of interest (ROI), 600 frames were recorded with an exposure time of 100 ms. A 640-nm excitation laser was used at 0.9 mW/cm$^2$, and acquisitions were performed in 1-min intervals. Data analysis was carried out using Deep-LASI[54]. Fluorescence traces were extracted and categorized. Traces in the "bleached" category were further analyzed to identify single- or double-step photobleaching events.

## Orientation of TmrAB in liposomes

The membrane orientation of reconstituted TmrAB was determined by proteolytic cleavage of the C-terminal His$_{10}$-tag on TmrA using Tobacco Etch Virus (TEV) protease. Proteoliposomes (5 mg/mL) were incubated with 0.2 mg/mL TEV protease for 18 h at 8 °C, either in the absence or presence of 1% (v/v) Triton X-100 to differentiate externally versus internally oriented TmrAB populations. Samples were analyzed

by SDS-PAGE followed by immunoblotting against the $His_{10}$-tag to assess protease accessibility.

## Peptide synthesis

The 9mer peptides GIINTLEEL, RRYQKSTEL, and RRYC[FL]KSTEL ([FL], fluorescein) were synthesized on pre-loaded Fmoc-L-Leu resin using a Liberty microwave-assisted peptide synthesizer (CEM) following a standard protocol (54 W, 3 min, 75 °C). Each coupling step was performed twice using 0.2 M fluorenylmethoxycarbonyl (Fmoc)-protected amino acid, 0.5 M O-benzotriazole-N,N,N',N'-tetramethyluronium-hexafluoro-phosphate (HBTU), and 1-hydroxybenzotriazole hydrate (HOBt:$H_2O$). Fmoc deprotection was carried out with 20% (v/v) piperidine and 0.1 M HOBt*$H_2O$ in dimethylformamide (DMF). Peptides were cleaved from the resin using a cocktail of 92.5% (v/v) trifluoroacetic acid (TFA), 2.5% (v/v) $H_2O$, 4.5% (v/v) thioanisole, and 0.5% (v/v) 1,2-ethanedithiol (EDT) for 1.5 h at room temperature. The cleaved peptides were precipitated in ice-cold diethyl ether (Et$_2$O), pelleted, dissolved in tert-butanol (tBuOH)/water (4:1, v/v), and lyophilized. Peptides were purified by reverse-phase (RP)-$C_{18}$ HPLC (PerfectSil C18 column; MZ-Analysentechnik) using buffer A (Milli-Q water with 0.05% (v/v) TFA) and buffer B (acetonitrile with 0.05% (v/v) TFA). Peptide identity and purity were confirmed by LC-MS. For fluorescence labeling, the peptide RRYCKSTEL was dissolved in PBS (10 mM $Na_2HPO_4$, 1.8 mM $KH_2PO_4$, 137 mM NaCl, 2.7 mM KCl, pH 7.4), supplemented with 3.5 mM DMF and incubated for 1 h at room temperature with a 1.3-fold molar excess of 5-iodoacetamido fluorescein (Merck).

## Mass spectrometry

All recombinantly produced proteins and chemically synthesized peptides were analyzed by liquid chromatography-coupled mass spectrometry (LC-MS) using a Waters BioCatrd system operating with UNIFY 3.1.0 software. Peptides were separated on an Acquity BEH C18 column (1.7 μm, 2.1 × 100 mm) using a 6.5-min linear gradient from 2 to 80% acetonitrile in Milli-Q water. Proteins were separated on an Acquity BEH C4 column (1.7 μm, 2.1 × 50 mm) using a 6.5 min linear gradient from 5 to 80% acetonitrile in Milli-Q water; both mobile phases were supplemented with 0.1% formic acid. Spectra were acquired in positive ion mode with a cone voltage of 30 V and a capillary voltage of 0.8 kV.

## LC-MS-based peptide transport assay

Peptide translocation by TmrAB[WT] was measured using LC-MS. Proteoliposomes (2 mg/mL final) were incubated at 45 °C for 45 min with RRYQKSTEL (50 μM or 200 μM), GIINTLEEL (50 μM), or the fluorescein-labeled peptide RRYC[FL]KSTEL (C4F, 5 μM) in the presence of 3 mM $MgCl_2$ and 3 mM ATP or ADP. All reactions contained an ATP regeneration system (0.5 mg/mL creatin kinase and 10 mM creatine phosphate) in liposome buffer. Reactions were stopped with ice-cold liposome buffer containing 10 mM EDTA. Samples were ultracentrifuged (450,000 × g, 30 min, 4 °C), and pellets were resuspended in 200 μL of 100 mM $Na_2CO_3$ (pH 11.5), followed by a second ultracentrifugation under the same conditions. Final pellets were resuspended in Milli-Q water and sonicated for 2 min. For quantification, C4F was spiked into samples containing RRYQKSTEL and GIINTLEEL as an internal standard. A final ultracentrifugation (450,000 × g, 1.5 h, 4 °C) was performed before LC-MS analysis using a Waters BioCatrd system (UNIFY 3.1.0). Peptides were separated on an Acquity BEH C18 column (1.7 μm, 2.1 × 50 mm) and analyzed in positive ion mode using a cone voltage of 30 V and capillary voltage of 0.8 kV. Quantification was based on total ion chromatogram peak areas at the peptide-specific retention times.

## Filter plate-based peptide transport assay

Peptide transport by TmrAB[WT] was quantified using the fluorescent peptide RRYC[FL]KSTEL (C4F) in a 96-well filter plate assay.

Proteoliposomes (1 mg/mL final) were incubated at 45 °C for 15 min with varying concentrations of C4F, 3 mM $MgCl_2$, and 3 mM ATP or ADP in liposome buffer. Transport was terminated by the addition of ice-cold liposome buffer containing 10 mM EDTA. Samples were transferred to polyethyleneimine (PEI)-precoated 96-well filter plates (Durapore membrane, 0.65 μm pore size, Millipore) and washed five times with 200 μL stop buffer to remove free peptides. Encapsulated C4F was released by incubating wells with PBS (pH 7.4) containing 0.1% (w/v) SDS for 10 min. The resulting lysates were transferred to black 96-well microtiter plates, and fluorescence was measured using a CLARIOstar plate reader (BMG LABTECH, Clariostar5.20 R5, Mars 3.10 R5) at excitation/emission wavelengths of 485/520 nm.

## Single-liposome transport monitored by flow cytometry

Single-liposome transport was quantified using a flow cytometry-based assay as previously described[33]. Proteoliposomes containing TmrAB[WT] were labeled with AF555-conjugated nanobody (Nb9F10[S63C-AF555]) at a TmrAB/nanobody molar ratio of 1:2. Labeled proteoliposomes were washed twice by ultracentrifugation (270,000 × g, 30 min, 4 °C) and incubated with 30 μM C4F peptide, 5 mM $MgCl_2$, 3 mM ATP or ADP for 10 min at 45 °C. Following incubation, samples were washed twice using the same ultracentrifugation conditions to remove unbound peptide. Mean fluorescence intensities corresponding to nanobody-tagged TmrAB and transported C4F peptide were measured using a FACS Celesta flow cytometer (BD Biosciences, BD FACSDiva 8.0.1.1). A total of $10^5$ proteoliposomes were recorded per sample based on forward scatter (FSC) and side scatter parameters. Single-liposome events were gated by FSC height versus area, and AF555-positive events were further gated to selectively analyze nanobody-decorated proteoliposomes. Data were processed using FlowJo software (v10.6.1). All measurements were performed in triplicates, and results are presented as mean ± s.d.

## Fluorescence anisotropy

Fluorescence anisotropy measurements were performed to assess the dynamics of AF555- and AF647-labeled OppA. Experiments were conducted in buffer containing 25 mM HEPES-NaOH pH 7.5, 150 mM NaCl. Labeled OppA was mixed with liposomes at a 1:1 (w/w) protein-to-liposome ratio. Measurements were performed using a CLARIOstar plate reader (BMG LABTECH) with excitation/emission settings of 482/535 nm and a long-pass dichroic filter LP 504.

## Time-correlated single-photon counting (TCSPC)

Fluorescence lifetime measurements were performed using a FluoTime 100 spectrometer (PicoQuant) equipped for time-correlated single-photon counting. Experiments were conducted in 25 mM HEPES-NaOH, pH 7.5, 150 mM NaCl, and included free dye, OppA, and liposome-encapsulated OppA samples. AF555 and AF647 were excited at 510 nm and 610 nm, respectively. Emission was detected through a 620/60 nm bandpass filter for AF555 and a BG4 700 nm long-pass filter for AF647. Photon arrival times were recorded until the TCSPC histogram reached a peak of 10,000 photons. Fluorescence decay curves were analyzed by fitting mono-or biexponential decay models using FluoFit (PicoQuant).

## Functionalization of glass slides for single-molecule FRET analysis

Glass coverslides were functionalized via aminosilanization as previously described[55]. Briefly, coverslides were cleaned by sequential sonication in Milli-Q water and analytical-grade acetone (>99.9%), followed by oxygen plasma treatment (0.3 mbar, 80% power, 15 min), and a 10 min incubation in methanol (100 mL). Subsequently, coverslides were incubated for 20 min in a silanization solution containing 100 mL methanol (analytical grade), 5 mL acetic acid, and 3 mL 3-aminopropyltrimethoxysilane (APTES). After brief sonication (1 min),

the slides were incubated for an additional 10 min, rinsed four times with methanol, and dried under a stream of nitrogen gas. To functionalize the surface with polyethylene glycol (PEG), a mixture of biotinylated-PEG (4% v/v) and non-biotinylated PEG was coupled to the amino groups via N-hydroxysuccinimide (NHS) chemistry. Specifically, 3.5 mg of biotin-PEG-NHS (5 kDa) were dissolved in 750 μL of 0.1 M $NaHCO_3$ PEGylation-buffer (pH 8.5), followed by the addition of 85 mg $CH_3O$-PEG-NH-CO-$C_2H_4$-CONHS (5 kDa). Air bubbles were removed by centrifugation at $16,000 \times g$ for 1 min. The PEG solution was sandwiched between two coverslips and incubated overnight in a humidity chamber. Coverslides were then rinsed thoroughly with Milli-Q water, dried with nitrogen gas, and subject to a second PEGylation step using $CH_3$-PEG-NHS (333 Da) in PEGylation-buffer (25 mM final) in a similar sandwich configuration. This second PEGylation was performed overnight under humid conditions. Finally, slides were rinsed with Milli-Q water, dried under nitrogen gas, and stored at −20 °C under argon gas until use.

## Liposome size distribution on the microscope surface

Liposomes containing single TmrAB[WT] were stained with the lipophilic carbocyanine dye DiD (0.1 mol%) and tethered to the glass surface via biotinylated PEG$_{11}$-conjugated nanobody Nb9F10. Fluorescence intensities were quantified by dSTORM, integrating the signal within a circular region of constant area surrounding each liposome. The resulting integrated fluorescence intensity per area was converted into liposome diameters by calibrating the NanoImager S (Oxford Nanoimaging) scale bar against TetraSpeck™ calibration beads (Thermo Fisher Scientific). The fluorescence signal from a 100 nm liposome was used as a reference to convert the fluorescence intensities into liposome diameters.

## Theoretical calculation of a single peptide encapsulated inside liposomes

The concentration corresponding to a single peptide encapsulated in a spherical liposome was calculated as $c = n/V$, where $n = 1/N_A$ (Avogadro constant) and $V = 4/3\pi r^3$ is the liposome volume.

## Single-molecule FRET imaging

Single-molecule FRET experiments were conducted using a flow chamber system to immobilize liposomes and transporters in an uptake-competent orientation. Flow chambers were assembled by placing a biotin-PEG-functionalized glass slide (4% (v/v) biotin-PEG) onto a sticky-slide flow channel (I Luer 0.8 mm, Ibidi), with the functionalized surface facing inward. The flow chamber was connected via silicon tubing, and flow was controlled using a 20 mL syringe. The chamber was flushed with 1 mL liposome buffer, followed by incubation with streptavidin (0.2 mg/mL) for 30 min at 8 °C to enable binding to the biotin-PEG surface. After washing with 1 mL liposome buffer, the surface was functionalized by incubating with either biotinylated anti-His antibody (0.02 mg/mL) (Abcam) or biotinylated-PEG$_{11}$-Nb9F10$^{S63C}$ (0.3 mg/mL) for 30 min at 4 °C, followed by an additional rinse with 1 mL liposome buffer. For anti-His antibody tethering, 100 nM fluorophore-labeled OppA was introduced into the flow chamber. For nanobody-based tethering, proteoliposomes (5 mg/mL) were centrifuged at $2000 \times g$ for 2 min at 4 °C and subsequently incubated on the surface for 30 min at 8 °C. Unbound material was removed by five successive washes with 1 mL liposome buffer, followed by 1 mL imaging buffer (25 mM HEPES-NaOH, pH 7.5, 150 mM NaCl, 5% (v/v) glycerol, 90 mM glucose, 5 mM Trolox, pyranose oxidase (7.5 U/mL), and catalase (1 kU/mL)). For the experiments involving valinomycin, 10 mM KCl was added to the imaging buffer. For single-translocation events with TmrA[EQ]B, proteoliposomes were incubated with pre-heated imaging buffer at 45 °C for 5 min followed by incubation with 200 μM RRYQKSTEL peptide and 3 mM ATP in pre-heated imaging buffer at 45 °C for 5 min. Subsequently, excess peptide and ATP

were removed by washing with 1 mL cold liposome buffer and 0.7 mL imaging buffer. The second translocation event was initiated by following the same procedure. For ATP-EDTA experiments, liposomes were pre-incubated with 10 mM EDTA for 5 min to chelate divalent cations. Imaging was performed using alternating laser excitation (ALEX) on a TIRF microscope (NanoImager S, Oxford Nanoimaging). Typically, 600 frames were acquired per ROI using an exposure time of 100 ms. Laser powers were set to 0.8 mW/cm² (532 nm excitation) and 0.9 mW/cm² (640 nm excitation), with data recorded in 1-min intervals.

## Single-molecule FRET data analysis

Single-molecule FRET (smFRET) measurements were performed using ALEX, allowing the assignment of detected photons based on both excitation and emission wavelength. Photon counts were classified into three detection channels: donor excitation with donor emission ($F_{DD}$), donor excitation with acceptor emission ($F_{DA}$), and acceptor excitation with acceptor emission ($F_{AA}$). From these signals, FRET efficiency ($E$) and stoichiometry ($S$) were calculated and FRET traces extracted using NanoImager software (ONI Nanoimager Development Build) and analyzed with deepFRET[56]. An 80% confidence threshold was applied for automated trace selection. Following automated classification, traces were manually curated to ensure data quality. Dynamic smFRET traces were further analyzed via HMM by deepFRET. One-dimensional histograms of $E$ and $S$ values were generated and visualized using OriginPro 2024 (OriginLab) and fitted with Gaussian distributions to identify distinct FRET populations.

## ATPase activity

ATP hydrolysis by TmrA[EQ]B and TmrAB[WT] was quantified using radiolabeled ATP. TmrAB reconstituted into liposomes (0.2 mg/mL) at a protein-to-lipid ratio of 1:20 (w/w) were incubated with 2 mM ouabain, 10 mM NaN₃, 50 μM EGTA, and 0.3 mM ATP spiked with [γ$^{32}$P]ATP (Hartmann Analytic), supplemented with either 5 mM MgCl$_2$ or 10 mM EDTA. Reactions were incubated for 18 min at 45 °C with or without 200 μM RRYQKSTEL peptide. Non-specific background hydrolysis was assessed in parallel for reactions lacking TmrAB. After incubation, aliquots were spotted onto polyethyleneimine cellulose thin-layer chromatography (TLC) plates (Merck Millipore). TLC separation was performed using 0.8 M LiCl in 0.8 M acetic acid as the mobile phase. Plates were dried, exposed overnight in an Exposure Cassette-K (Bio-Rad), and radiolabeled products were visualized using a Personal Molecular Imager System (Bio-Rad).

## Reconstitution in lipid nanodiscs

Reconstitution of wild-type TmrAB (TmrAB[WT]) and the slow-turnover variant TmrA[EQ]B into lipid nanodiscs was performed as previously described[14]. Briefly, bovine brain lipids (Sigma-Aldrich) were solubilized in 50 mM HEPES, pH 7.4, 150 mM NaCl supplemented with 20 mM β-DDM at 37 °C for 30 min. Purified TmrAB, MSP1D1 scaffold protein, and solubilized lipids were mixed at a molar ratio of 1:7.5:100 (TmrAB:MSP1D1:lipid) in TmrAB-SEC buffer lacking detergent. The final lipid-to-detergent ratio was 1:1.4 (w/w). The mixture was incubated at 20 °C for 30 min before initiating nanodisc self-assembly by sequential addition of Bio-Beads SM-2 (Bio-Rad) at 8 °C: 40 mg/mL for 1 h, followed by 80 mg/mL overnight to remove detergent. Following detergent removal, nanodisc-containing samples were concentrated using Amicon Ultra centrifugal filters (0.5 mL, 100 kDa MWCO; Merck Millipore). To label the complexes, reconstituted TmrAB-nanodisc complexes were incubated with the nanobody Nb9F10$^{S63C}$ at a 1:1 molar ratio for 10 min at 20 °C. Subsequently, samples were analyzed by size-exclusion chromatography (SEC) using a Superdex 200 Increase 3.2/300 column (Cytiva) equilibrated in TmrAB-SEC buffer without detergent to separate nanobody-bound TmrAB containing nanodiscs from empty nanodiscs.

## Cryo-EM sample preparation and data acquisition

Nanodiscs-reconstituted TmrAB^WT and TmrA^EQB bound to nanobody were incubated with 10 mM ATP, 10 mM EDTA, and 200 µM peptide RRYQKSTEL for 5 min at 45 °C. Immediately thereafter, 3 µL of each sample was applied to freshly glow-discharged Quantifoil grids (R1.2/1.3, 300 mesh), blotted, and plunge-frozen in liquid ethane using a Vitrobot Mark IV (Thermo Fisher Scientific) operated at 4 °C and 100% humidity. Cryo-EM data were collected on a 200 kV Glacios transmission electron microscope (Thermo Fisher Scientific, EPU 3.10.0) equipped with a Falcon 3EC direct electron detector (Thermo Fisher Scientific) operated in counting mode. Micrographs were acquired at a nominal magnification of 150,000×, corresponding to a calibrated pixel size of 0.95 Å. Dose-fractionated movies were recorded at an electron flux of 0.8 e⁻ pixel⁻¹ s⁻¹ over a total exposure time of 33.35 s distributed over 24 frames, resulting in a cumulative dose of 28.3 e⁻ Å⁻² (see Supplementary Table 1).

## Cryo-EM image processing

Cryo-EM data were processed using cryoSPARC v4.6.0–4.7.0 (ref. 57), following a common workflow for both TmrAB^WT and TmrA^EQB, as illustrated in Supplementary Figs. 12 and 13. After Patch Motion Correction and Patch CTF estimation, micrographs were curated by selecting those with CTF fits better than 5 Å for downstream processing. Particle picking was performed using TOPAZ 0.2.5a (ref. 58), followed by particle extraction with three-fold Fourier cropping (2.85 Å pixel⁻¹). Extracted particles were subjected to multi-class ab initio reconstruction, and classes displaying a feature characteristic of type IV ABC transporters were selected for iterative heterogeneous refinement to enrich for high-quality TmrAB particles. Selected particles were then re-extracted at full pixel size (0.95 Å pixel⁻¹) and subjected to non-uniform refinement, yielding maps at 3.26 and 3.20 Å resolution for TmrAB^WT and TmrA^EQB, respectively, based on the gold-standard Fourier shell correlation (FSC = 0.143) criterion. To further improve resolution, reference-based motion correction and CTF refinement were applied, followed by a final round of non-uniform refinement. This resulted in final reconstructions at 3.02 Å resolution for both TmrAB^WT and TmrA^EQB.

## Molecular modeling

The previously determined structure of TmrA^EQB in OF occluded conformation (PDB: 6RAI) was used as a starting model and docked into the corresponding cryo-EM maps using UCSF ChimeraX 1.9 (ref. 59). For modeling TmrAB^WT, residue 523 was reverted from glutamine to the native glutamic acid. Both TmrAB^WT and TmrA^EQB models were independently refined in real space against their respective cryo-EM density maps using Coot 0.9.8.95 (ref. 60) for manual model adjustment and Phenix 1.21.2-5419 (ref. 61) for automated refinement.

## Statistical analysis

Statistical significance analysis was performed using GraphPad Prism 8.0.2. ATPase data were analyzed by two-way ANOVA. Statistical tests and $P$ values are reported in the figure legends. For TmrAB^WT, the $F$ value for the comparison of MgCl$_2$ versus EDTA without the peptide is 337.7, and with the peptide, 338.8, with a degree of freedom of df$_{between}$ = 1 and df$_{within}$ = 4. For TmrA^EQB, the $F$ value for the comparison of MgCl$_2$ versus EDTA without the peptide is 268, and with the peptide, 87.7, with a degree of freedom of df$_{between}$ = 1 and df$_{within}$ = 4.

## Reporting summary

Further information on research design is available in the Nature Portfolio Reporting Summary linked to this article.

## Data availability

The cryo-EM maps of the wild-type TmrAB and the slow-turnover variant TmrA^EQB in the outward-facing occluded, ATP-bound state in the absence of Mg²⁺ have been deposited in the Electron Microscopy Data Bank under accession numbers: EMD-54378 for TmrAB^WT and EMD-54377 for TmrA^EQB. Atomic coordinates for the atomic models have been deposited in the Protein Data Bank [http://www.rcsb.org] under accession numbers PDB ID: 9RYF for TmrAB^WT and 9RYE for TmrA^EQB. All data are available in the main text or the Supplementary Information. All other data are available from the corresponding author upon request. Source data are provided with this paper within the Source data File. Source data is also available at [https://doi.org/10.25716/gude.0frp-685b]. Source data are provided with this paper.

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

## Acknowledgements

This work was supported by the European Research Council (ERC Advanced Grant 101141396 to R.T.) and the German Research Foundation via the Collaborative Research Center CRC 1507 (P18 and Cryo-EM Infrastructure Z02 to R.T.). The authors thank Bert Poolman's lab (University of Groningen, N.L.) for providing the LlOppA plasmid. The authors acknowledge Jan F.M. Stuke and Jonas Göhmann for their assistance in automating trace extraction from ONI NanoImager software, and Anastasia Kinzl for performing the immunoblotting and preparing the TLC plates. The authors are grateful to Dr. David Glück for the

TCSP measurements, feedback on the manuscript, and assistance with data analysis. The authors thank the Wachtveitl lab (Goethe University Frankfurt) for access to their FluoTime 100 spectrometer (PicoQuant), as well as the Heileman lab (Goethe University Frankfurt) for access to their plasma cleaner. The authors also thank Maximilian Zehetmaier for assistance with cartoon design and providing MSP1D1 nanodiscs, Dr. Rupert Abele for support with radioactivity experiments, and Matthias Rose and the Volker Müller lab (Goethe University Frankfurt) for access to the radioactivity lab. Finally, the authors thank Dr. Yudhajeet Basak, Inga Nold, and Andrea Pott for manuscript comments and proofreading.

## Author contributions

C.N. performed all single-molecule FRET experiments and data analysis, as well as *Ll*OppA purification, ATPase, LC-MS transport experiments, and provided the cryo-EM samples. T.N. and M.P. prepared all TmrAB samples and nanobodies, and carried out the nanobody binding and transport studies. L.S. carried out the single-particle cryo-EM analyses. A.F. performed the LC-MS measurements and data analysis. C.N. and R.T. wrote the manuscript. R.T. conceived and supervised the work.

## Funding

## Competing interests

The authors declare no competing interests.
