## [Transparent Peer Review file · Nature Communications]

Single-molecule dynamics reveal ATP binding alone powers substrate translocation by an ABC transporter

Corresponding Author: Professor Robert Tampé

Version 0:

Reviewer comments:

Reviewer #1

(Remarks to the Author)

see pdf - Recommendation is to work on and resubmit the paper with major revisions.

Reviewer #2

(Remarks to the Author)

The manuscript by Nocker is probing the role of Mg²⁺ in either coordinating ATP or facilitating hydrolysis and subsequent substrate transport of the ABC transporter TmrAB. The authors use a smFRET approach of TmrAB reconstituted in liposomes to monitor peptide transport by utilising the OppA as reporter. They evaluated their system in the presence of ATP-Mg²⁺ and peptide, with wt or ATPase deficient TmrAB EQ mutant. They also pretreated their proteoliposomes with EDTA and added ATP-EDTA, and they observed that a single peptide transport event takes place. Based on these observations they conclude that ATP alone is capable to drive transport but not resetting the transporter for a subsequent transport event. Finally, using cryo-EM analysis of TmrAB with ATP-EDTA, they show that the transporter is capable of adopting the outward facing conformation without Mg²⁺ further validating their findings.

The work is built on the authors expertise to study TmrAB. The experiments have been executed to high standards and with fairly sound conclusions. Some of the results such as the transport of single peptide is just a reconfirmation of their previous work about TmrAB transporting a single peptide per transport event. Debugging the role of Mg²⁺ is the novelty of this study.

My main concern is the lack of discussion on the presence of OppA species that do not show high FRET upon a transport event in all their experiments; under all conditions with ATP and peptide, the plots show a bimodal behaviour that the authors have not addressed and it presents a limitation of using OppA as the reporter rather than labelling TmrAB directly. This is the biggest weakness of this manuscript. The plots show a significant portion of OppA molecules not binding the peptide even under conditions of saturation. This could either be due to the presence of more than one TmrAB in the liposome (there is no evidence for this) or encapsulation of more than one OppA or a more complex transport kinetic model that needs further investigation.

The legend for Fig 3 f on 'transition kinetics' are not relevant to TmrAB as the FRET is only reporting on OppA binding kinetics which is not the focus of this study. This should be changed.

Fig2 would also benefit from having a HMM analysis table as Fig3, and make comparisons between the wt and mutant protein in the discussion.

Under the cryo-EM results section, the manuscript would benefit by providing a comparison of the NBDs with and without Mg²⁺; ie do the side chains adopt different rotamers? They should also provide rmsd values for the two structures. Are there any conformational changes within the NBD/coupling helix region that the absence of Mg²⁺ can induce?

The statement in page 10 'Yet our work uncovers a previously overlooked Mg²⁺- independent ATP-bound state that

supports single-turnover transport.' is confused with their previous sentence and there is not much relevancy for transporters which are greatly uncoupled for ATP-dependent substrate transport.

Supplementary Table 1 is reporting the EQ mutant structure but there is no figure of its ATP binding site as for the wt and any discussion beyond the difference maps. Supplementary Fig 8 should show the process of both wt and EQ mutant separately.

Reviewer #3

(Remarks to the Author)

Single-molecule dynamics reveal ATP binding alone powers substrate translocation by an ABC transporter.

The authors show that binding of ATP is sufficient for a half turnover of the peptide ABC transporter TmrAB, i.e. the transport of a peptide from cis to trans. Mg-ATP is required for hydrolysis and the other half turnover, which resets the system for the next translocation event. Thus, the conformational switch from inward-facing to outward-facing by ATP (without Mg²⁺ ions) is a key finding of this paper, which has been postulated before but may not have demonstrated as convincingly as done for TmrAB in this paper. Perhaps, the authors should summarize in their discussion to what extent the ATP-driven half turnover is unique for TmrAB (or heterodimeric ABCs), and indicate other studies where this has been shown directly or indirectly or ruled out. Now only computational studies and CFTR (special case) are mentioned in the Discussion section.

Overall, this is a beautifully executed single-molecule study on an important bacterial homologue of the human TAP system. The work is important and well presented. We have a few questions and minor comments for the authors to consider:

1. Figure 1d reports $K_d=0.9$ μ M (text indicates values from 0.2-0.9 μ M) for RRYQKSTEL binding to OppA; Figure 1f reports k_{on} and τ , from which the K_d can be calculated, which then yields $K_d=7\mu$ M. What is the explanation for the approximately 10-fold difference in binding constant?
2. The internal volume of the vesicles is calculated for a diameter of 160 nm, whereas the peak in the vesicle concentration is around 200 (Fig. 3g). Why is 160 nm chosen, and how does the distribution of vesicles sizes (from 100 to 500 nm) affect the analysis of the data (e.g. in the context of max 1 OppA and 1 TmrAB per vesicle, and both proteins need to be in the same vesicle)?
3. SFig.4 shows that nanobody binding does not affect the activity of TmrAB. Since a high protein-to-lipid ratio is used, what is the evidence that the reported activity is actually transport and not peptide binding? The same applies to the data of SFig.6. It should be possible to discriminate binding from transport by comparing the number of peptides translocated with the number of TmrAB molecules in the membrane.
4. Do non-hydrolyzable ATP analogues also trigger the half turnover of TmrAB?
5. How was OppA encapsulated in the vesicle lumen? Page 23 only describes the membrane reconstitution of TmrAB.
6. Figure 2ab show a high FRET peak, which is not seen in 3ab. Is this endogenously bound peptide in this particular preparation of OppA?
7. Figure 2c shows the time dependence of peptide translocation (0-10 and 11-20 min) for wildtype TmrAB, but this is not shown for the EQ mutant in Figure 3c or 3d. Shouldn't it be an order of magnitude slower?
8. Figure 6 suggests that 1 ATP is hydrolyzed per peptide translocated. Is there evidence for this contention; they also write on page 10 that wildtype TmrAB exhibits futile hydrolysis of ATP.
9. Page 28: It is not entirely clear how FRET traces were selected for the FRET distributions. Were traces without dynamics discarded? What were the criteria?
10. SFig. 3ab: Why a double peak in the SEC profile for wildtype TmrAB, which is not seen for the EQ protein.
11. SFig. 7a: The high FRET state after one translocation event is similar to the high FRET state after two translocation events of Fig. 3d. Similarly, in SFig. 10e, the high/low FRET ratio is lower after translocation event 2 than after 1 (and even after 3 events it may still be lower than after 1). A brief description on variations between experiments and reproducibility or other explanation would be helpful.
12. The description of the legend of SFig. 10g (Hidden Markov Model) is ambiguous and needs some rewriting.

Version 1:

Reviewer comments:

Reviewer #2

(Remarks to the Author)

The authors have addressed all my queries and the clarity of the revised manuscript has improved.

Reviewer #3

(Remarks to the Author)

I am impressed by the thoroughness of the rebuttal and the way the authors have addressed the comments and suggestion of the three reviewers. The rebuttal is a valuable addition to the manuscript. I have no further comments.

Reviewer #4

(Remarks to the Author)

See Attached PDF

Point-to-Point Response to Reviewers (NCOMMS-25-68980-T):

Reviewer #1 (Remarks to the Author)

ABC transporters represent a large family of primary active membrane transporters, which are conserved across all domains of life. They utilize ATP binding and hydrolysis energy to transport diverse substrates against concentration gradients across cellular membranes. Their involvement in human diseases, including multidrug resistance and genetic disorders, makes understanding their molecular mechanisms relevant. ABC proteins are categorized based on their nucleotide-binding domains (NBDs) and transmembrane domains (TMDs). While structural studies provided detailed insights into structural aspects of the transport mechanisms, single-molecule techniques have begun to overcome limitations of ensemble methods, providing insights into transport dynamics and revealing novel mechanistic details of substrate translocation events. In the submitted paper, a novel single-molecule platform combining FRET-based peptide sensing with conformationally selective nanobodies aims at direct observation of individual ABC transporter substrate translocation events.

The studied system was TmrAB a heterodimeric ABC transporter that mediates peptide translocation across bacterial membranes, functioning as a structural and functional homolog of the human TAP complex. Using a slow-turnover TmrAB variant (TmrA^{EQB}), the authors claim to observe discrete transport events at the single transporter level. Their interpretation is that ATP binding alone, even without Mg²⁺, can drive substrate translocation through an IF-to-OF conformational switch. These findings support a mechanistic model where ATP binding induces NBD dimerization and conformational switching, with peptide translocation occurring concurrently, followed by Mg²⁺-dependent ATP hydrolysis for transporter resetting. The authors suggest that their work establishes a single-molecule framework for understanding ABC transporter mechanisms, though limitations include temperature constraints for wild-type TmrAB studies and sensor saturation after few uptake events.

I find the topic and the approach of the paper timely, innovative, and very interesting, and in conclusion worthwhile to publish. The study was in parts done carefully, yet various conclusions and the main claim of the paper are not supported by the supplied data and the modelling. I thus render the paper in principle suitable for publication in Nature Communications due to its novelty and ideas, subject to the following major revisions that require additional experiments/data and analysis:

*Reply: We sincerely thank Reviewer #1 for the thoughtful and thorough evaluation of our manuscript. We are very grateful for the positive feedback regarding the **timeliness, innovation, and overall interest of our study**. We are encouraged by the assessment that our work is, in principle, **suitable for publication in Nature Communications** due to its **novelty and conceptual contribution**. We also appreciate the constructive feedback pointing out areas where additional experiments, data, and analysis are needed. We have carefully addressed all concerns raised, which we believe considerably strengthen both the robustness and clarity of our mechanistic conclusions.*

Specific Comments:

- 1) The paper lacks a summary of smFRET work on ABC transporters from various groups in which the major mechanistic conclusions should be summarized and the advances made (also in comparison to the current approach) are compared in a fair way

*Reply: We thank the reviewer for this important suggestion. We have now expanded the introduction to include a concise summary of previous smFRET studies on ABC transporter and their mechanistic insights. This addition also clarifies how our approach complements and advances beyond these earlier works. The **New Paragraph (page 3)** reads:*

“In particular, single-molecule Förster resonance energy transfer (smFRET) studies confirmed an alternating-access mechanisms in ABC transporters; for example, the bacterial homodimeric exporter McjD requires both substrate and ATP to adopt the outward-facing conformation

(Husada et al, 2018). Moreover, combining smFRET with single-particle cryo-EM has enabled visualization of the ABC transporter MRP1 under turnover conditions (Wang et al, 2020). Conformational changes in substrate-binding proteins (SBP) from bacterial ABC importers were investigated by smFRET (de Boer et al, 2019; Husada et al, 2015). The SBPs were repurposed as smFRET sensors to track individual translocation events in secondary active transporters, providing quantitative amino acid transport rates (Ciftci et al, 2020; Fitzgerald et al, 2019).“

This addition summarizes prior single-molecule approaches, acknowledges their contributions, and positions our work within the broader context of smFRET-based ABC transporter research.

- 2) Relevant references in which single-transporter assays were introduced are not mentioned (Stamou group and others); these should be integrated into the introduction and the framing of the paper.

Reply: *We appreciate the reviewer pointing out this omission. Although the Stamou’s work was already included in the original manuscript, it was not cited in the introduction. We now reference these key methodological contributions directly in the introduction:*

“Single-molecule techniques overcome ensemble averaging and provide high-resolution insights into transport dynamics (Blanco & Walter, 2010; Holden et al, 2010; Kosmidis et al, 2022).

The work is also discussed in the Discussion (Kosmidis et al, 2022), where we compare it to the “resting” state observed in our system. This strengthens the framing and situates our study within the broader landscape of single-transporter methods.

- 3) The capabilities of the developed assay should be compared against other format (current ref. 26-27), which allow real-time recordings. What are the pros and cons of the approach introduced?

Reply: *We thank the reviewer for this useful suggestion. We have expanded the Discussion to explicitly compare our assays with real-time assay formats (Ciftci et al, 2020; Fitzgerald et al, 2019), highlighting advantages and limitations.*

In brief, our assay offers several advantages:

- (i) Selective immobilization of uptake-competent transporters via a conformationally non-selective nanobody.*
- (ii) Direct observation of peptide translocation through an ABC transporter at the single-molecule level.*

These features allow mechanistic dissection of transport steps that remain hidden in ensemble assays.

The main limitations, now stated more explicitly, include:

- (i) Inability to perform single-molecule assays on wild-type TmrAB under fully physiological conditions (>45 °C) due to instrument and sensor thermostability constraints.*
- (ii) Sensor saturation after a few uptake events, which currently precludes determining whether transport cycles proceed in bursts or continuously.*

This expanded discussion highlights the complementary nature of our assays relative to existing real-time formats and provides a balanced view of its strengths and constraints.

- 4) **Figure 1**; please add supplementary data of traces of all conditions with at least 20-30 traces per condition.

***Reply:** We thank the reviewer for this suggestion. We have now added representative single-molecule FRET traces for all conditions in the **New Supplementary Figs. 3 and 4**, providing comprehensive visualization of the single-molecule behavior across all experimental conditions.*

- 5) **Figure 1d**; analysis of data is not correct or not correctly described. Normalized acceptor fluorescence is not identical to the area of the “blue” closed-state population shown in panels c; it would have to be the normalized area of one against the other population. According to panel c, there is saturation at closed-state populations of 60% so the panel needs attention

***Reply:** We appreciate the reviewer’s accurate observation. We have corrected the legend and text associated with Fig. 1d to clearly distinguish between normalized acceptor fluorescence and the population areas derived from Gaussian fits in Fig. 1c. This removes the previous ambiguity.*

- 6) **Figure 1c**; please explain why the sensor has the limitation that only 60% are active or why the sensor does not saturate in the fully closed state? The problem persists in liposomes (Figure S2) but here also the apo state shows a substantial proportion of closed state in the absence of peptide. Maybe analysis of traces from individual liposomes can help? Please comment and explain.

***Reply:** We thank the reviewer for raising this important point. The apparent plateau at ~60% does not reflect intrinsic limitation of the sensor. Instead, it arises from constraints in Gaussian deconvolution when fitting small subpopulations in FRET histograms. As a result, the high-FRET fraction is underestimated, whereas inspection of the raw, higher-resolution histogram shows a larger proportion of molecules in the closed/high-FRET state. A similar underestimation was also observed in our initial liposome measurements. We have now clarified this in the manuscript and added a brief explanation in the figure legend.*

As outlined in our response to Reviewer #2 (comment 1), the histograms represent a mixture of low- and high-FRET OppA populations. Peptides are released after 0.5 s resulting in a shift from high to low FRET and the peptide concentration determines how fast the reassociation happens. A small low FRET population will be always present and achieving complete sensor saturation requires multiple peptide translocation events and very high peptide concentrations. To directly demonstrate this, we performed an additional experiment in which three success translocation events mediated by TmrA^{EQB} resulted in full sensor saturation (Fig. R2).

*Importantly, after optimizing OppA purification and sample handling, we were able to eliminate nearly all of the high-FRET apo population. The **New Supplementary Fig. 17** (AMP-PNP control) confirms a clean low-FRET baseline in the absence of peptide.*

Finally, we note that our HMM analysis is based solely on dynamic, single-liposome trajectories and is not affected by these histogram-fitting artifacts.

- 7) **Figure 1e,f**; please provide the dwell-time histograms of the different conditions used to calculate rates in panel f; to me the trace shown in e, which I assume is close to K_D , should have a closed-state lifetime longer than 500 ms; more data/analysis is needed to give confidence in the analysis

***Reply:** We thank the reviewer for emphasizing the importance of dwell-time analyses. As noted, individual OppA molecules exhibit considerable heterogeneity in closed-state lifetimes, even within the same smFRET trajectory. Some closing events last only a few milliseconds, while*

others persist for seconds. This variability reflects the intrinsic stochastic dynamics of OppA and illustrates why single-molecule measurements are essential to resolve behaviors that are averaged out in ensemble analyses.

The rates reported in Fig. 1f are based on a large dataset of transitions ($n = 1,311$), collected from multiple molecules across all conditions. Although individual traces – such as the example shown in Fig. 1e – may include events with closed-state lifetimes longer than 500 ms (or shorter), these fluctuations are fully incorporated into the overall statistical analysis. Comparable heterogeneity in OppA closed-state dwell times has been reported previously; for example, (de Boer et al, 2019) (Supplementary Fig. 2-1) observed variations of up to a factor of three within a single smFRET trace at peptide concentration near the K_D .

In response to the reviewer's request, we now provide dwell-time histograms for all experimental conditions used to derive the rates in Fig. 1f. These are included as Fig R1. The histograms summarize open- and closed-state dwell times under each experimental condition.

We also note that temporal resolution is inherently limited by the acquisition speed of the microscope. Within this constraint, calculated closing rates show a clear linear dependence on peptide concentration. This relationship enables us to relate OppA closing kinetics to luminal peptide concentrations, which are in excellent agreement with theoretical estimates.

Fig. R1: Dwell time histograms of OppA at different peptide concentrations. **a**, Open-state dwell-time distributions showing progressively shorter open-state lifetimes with increasing concentrations of peptide (RRYQKSTEL). **b**, Closed-state dwell-time distributions demonstrating consistent closed-state lifetime across all peptide concentrations.

- 8) The above problems persist in Figures 2-3 and pose a challenge to interpret data in figure 3. How do the distributions of open/closed state of individual liposomes look like? If the main claim to make is that 1 or 2 peptides get transported than it would be good to analyze the data accordingly and show that in some liposome the on/off-rate or FRET histogram support this while in other maybe not transport has occurred; currently the interpretation and possibilities are averaged out in both figures and no data/analysis of individual liposomes is shown. I highly recommend to change this.

Reply: We fully agree with the reviewer that single-liposome-level analysis is essential for a correct interpretation of the data in Figure 2. By calibrating liposome size and correlating the fluorescence intensity of individual liposomes with FRET sensor responses, we can directly monitor and quantify peptide transport events on a per-liposome basis.

This refined analysis allows us to resolve discrete single-peptide transport steps and to distinguish liposomes that undergo one or two transport events from those that show no transport. Accordingly, the variability in open/close-state distributions across individual liposomes is now captured, rather than averaged out, and we show representative cases illustrating the expected heterogeneity.

A detailed description of the reconstitution strategy, imaging workflow, and per-liposome analysis is provided in our response to point 12 (Reviewer #1) and in the revised Methods section. These additions clarify the homogeneity of the preparation and demonstrate that the observed transport events are robust at the single-liposome level.

- 9) The liposome size distribution is heterogenous with a population of 200 nm diameter (40%), but already this one has a substantial shoulder with 250 nm diameter (40%) and another large one >300-350 nm (20%); thus, any concentration calculation from rates of blinking will be completely different for these liposome size (up to factors of 5-10!)

Reply: *We fully agree with the reviewer that liposome size heterogeneity can strongly influence concentration estimates. As detailed in our response to point 12 (Reviewer #1), we have now implemented improved liposome preparation and size characterization, followed by refined on-surface size calibration. These steps result in a substantially more homogeneous population of liposomes with diameters of ~100 nm diameter. This improved uniformity markedly reduces variation in the luminal volume and therefore minimizes differences in peptide concentration per liposome after transport.*

- 10) The estimation made in panel f to relate 4 μM to 1 and 6 μM to 2 peptides is incorrect and thus fundamental conclusions made in the paper are not supported/valid. The paper says “These luminal peptide concentrations matched theoretical estimates for approximately one and two peptides in liposomes ranging of 200 ± 10 nm in diameter (Fig. 3g), ...”. My own estimate of the calculations in excel below shows a fundamental problem of the idea of the authors, i.e., to distinguish 1 or 2 peptide molecules but also use a heterogenous mixture of liposomes for this. In other words – currently it is not possible (explained below).

Reply: *We acknowledge this limitation. To address it, we have added more detailed information on liposome reconstitution, centrifugation, and size calibration. This workflow produces a uniform population of ~100 nm diameter liposomes. With this well-defined size, the theoretical luminal concentration of a single peptide per liposome (~3.2 μM) aligns with the sensor's K_D value, allowing us to distinguish single- versus multiple-peptide transport events. These updated methods and corresponding data are presented in the **New Supplementary Fig S11** and included in Fig. 3g.*

- 11) Arising problem 1: the liposome size should be smaller so that the sensor range with K_D of 1 μM is most sensitive to changes of 1 to 2 peptides, for this a diameter of 150 nm would be ideal (see below)

Reply: *Following this suggestion, we optimized our liposome preparation to achieve a mean diameter of ~100 nm. At this size, the luminal peptide concentration resulting from single-peptide occupancy falls within the optimal detection range of the FRET sensor ($K_D \approx 1 \mu\text{M}$), thereby ensuring high sensitivity to single-peptide events while maintaining robust and reliable signal detection. This adjustment also minimizes variability due to liposome size heterogeneity.*

- 12) Problem 2: the liposome size distribution together with the averaging the authors do will not allow them to make any conclusion since the differences between liposomes of 200 nm and 350 nm has tremendous differences in the molecule number per vesicle (10 for 4 μ M for 200 nm and 54 for 350 nm); in principle the best solution would be to calibrate the liposome size in each image using an intensity-based size measure (e.g., with a lipid dye) relating the intensity distribution to the size distribution.

Reply: *We fully agree with this point and have implemented the following improvements:*

- (i) **Liposome size determination on the microscope surface:** *Liposomes were stained with the lipophilic dye DiD and immobilized on PEG-biotin surfaces via nanobody interactions to capture single TmrAB transporters in an uptake-competent orientation. Fluorescence intensity of individual liposomes was used to infer their diameters, confirming homogeneity and revealed that the majority of captured liposomes at the surface have a diameter of \sim 100 nm (New Fig. 3g and New Supplementary Fig. 11).*
- (ii) **Implications for peptide quantification:** *With this uniform liposome population, single-peptide concentrations correspond to \sim 3.2 μ M, enabling accurate analysis of single-transport events and eliminating artifacts from averaging across heterogeneous liposome sizes. These improvements ensure that the single-liposome FRET data reliably supports our conclusions regarding single-peptide transport events.*

- 13) The author also do not provide another absolutely essential control, which is related to their approach. Since they have a label-free approach (from the viewpoint of the transporter) they have no readout to see how many transporters per liposome (with varying size) actually caused the change of sensor read-out. What is needed: label the transporter and reconstitute it at the same protein-to-lipid ratio as in the normal assays and count how many transporters are in the liposome to understand this effect.

Reply: *We appreciate the reviewer's suggestion. To address this, we carefully controlled the protein-to-lipid ratio during reconstitution such that, statistically, only one transporter is present per \sim 10 liposomes. Consequently, the majority of liposomes are empty and removed during washing, leaving a small population containing a reconstituted TmrAB.*

Additionally, to directly assess transporter copy number per liposome, we labeled TmrAB^{WT} containing a single cysteine with AF647. Using the same reconstitution protocol, one-step and two-step photobleaching events were counted (New Supplementary Fig. 6). This analysis demonstrates that 95.6% of liposomes contain a single reconstituted transporter.

This stochastic reconstitution approach, combined with orientation-selective immobilization, ensures that the observed FRET responses predominantly arise from single transporters per liposome.

- 14) Another possibility would be to perform control experiments in which 1 or 2 peptides are reconstituted into the liposomes (without transporter) but with sensor to extract kinetics off/on.

Reply: *We thank the reviewer for this suggestion. However, due to the stochastic nature of encapsulation, it is not feasible to prepare liposome populations containing exactly one or two peptides without introducing substantial heterogeneity. In contrast, our TmrA^{EQB} single-turnover assay inherently restricts peptide transport to single events per liposome within the assay window, providing a well-defined system to accurately quantify single-peptide translocation.*

- 15) To say it clearly: currently the paper cannot support any claims and interpretations made further on the mechanistic side since the controls are lacking, and importantly: the calculations and assumptions on the ability of the assay to see single-peptide transport are not correct (and not provided in the manuscript, so it is difficult to see where the error comes from or if different assumptions were made).

*Reply: We thank the reviewer for this important comment. To address these concerns, we have included extensive additional controls, expanding the **Supplementary Information from Fig. 10 to Fig. 17**. Mechanistic interpretations based on single-molecule results are further corroborated by orthogonal approaches, including cryo-EM, radioactivity-based ATPase measurements, and ensemble transport assays based on LC-MS.*

Regarding the theoretical calculations of single-peptide encapsulation, the original calculations were correct. To enhance clarity and transparency, we have now explicitly provided these calculations in the Methods section. With the updated, well-defined size of surface-immobilized liposomes (~100 nm diameter), the theoretical luminal concentration of a single peptide aligns closely with the observed smFRET data. These improvements confirm that our experimental setup reliably reports single-peptide transport events, thereby supporting the mechanistic conclusions presented in the manuscript.

Reviewer #2 (Remarks to the Author)

The manuscript by Nocker is probing the role of Mg^{2+} in either coordinating ATP or facilitating hydrolysis and subsequent substrate transport of the ABC transporter TmrAB. The authors use a smFRET approach of TmrAB reconstituted in liposomes to monitor peptide transport by utilizing the OppA as reporter. They evaluated their system in the presence of ATP- Mg^{2+} and peptide, with wt or ATPase deficient TmrAB EQ mutant. They also pretreated their proteoliposomes with EDTA and added ATP-EDTA, and they observed that a single peptide transport event takes place. Based on these observations they conclude that ATP alone is capable to drive transport but not resetting the transporter for a subsequent transport event. Finally, using cryo-EM analysis of TmrAB with ATP-EDTA, they show that the transporter is capable of adopting the outward facing conformation without Mg^{2+} further validating their findings.

The work is built on the authors expertise to study TmrAB. The experiments have been executed to high standards and with fairly sound conclusions. Some of the results such as the transport of single peptide is just a reconfirmation of their previous work about TmrAB transporting a single peptide per transport event. Debugging the role of Mg^{2+} is the novelty of this study.

*Reply: We sincerely thank Reviewer #2 for the encouraging and thoughtful evaluation of our work. We are pleased that the reviewer recognizes the **rigor and high quality** of our experiments and finds our **conclusions sound**. We particularly appreciate the acknowledgement of our expertise with TmrAB and are pleased that the **novelty of our study**—clarifying the role of Mg^{2+} in ATP-driven transport—is recognized as an **important advancement** to understanding the mechanistic basis of single-peptide transport.*

Specific Comments:

- 1) My main concern is the lack of discussion on the presence of OppA species that do not show high FRET upon a transport event in all their experiments; under all conditions with ATP and peptide, the plots show a bimodal behavior that the authors have not addressed and it presents a limitation of using OppA as the reporter rather than labelling TmrAB directly. This is the biggest weakness of this manuscript. The plots show a significant portion of OppA molecules not binding the peptide even under conditions of saturation. This could either be due to the presence of more than one TmrAB in the liposome (there is no evidence for this) or

encapsulation of more than one OppA or a more complex transport kinetic model that needs further investigation.

Reply: We thank the reviewer for raising this important point. The observed histograms reflect both low- and high-FRET populations of OppA. Peptides are released after ~ 0.5 s, resulting in a shift from high to low FRET, and the peptide concentration determines the rate of reassociation. A small low FRET population is therefore always present, and achieving complete sensor saturation requires multiple peptide translocation events at very high peptide concentrations. In our experiments, typically only one or two translocation events occur per liposomes – insufficient to fully shift all OppA sensors into the high-FRET state. To demonstrate this, we performed an additional experiment with **three sequential translocation events** through TmrA^{EQB} (Fig. R2), which shows near-complete saturation of the sensor and a corresponding shift to the high FRET state.

A small fraction of OppA molecules remain static in the low-FRET state, likely due to damaged sensors or transporters. Importantly, our Hidden Markov Model (HMM) analysis excludes these static traces, considering only dynamic traces that exhibit transitions upon peptide binding. This ensures that the kinetic rates reported are derived solely from OppA molecules actively reporting transport events. By contrast, the histograms include both static and dynamic traces, which explains the persistent low-FRET populations.

While direct labeling of TmrAB would provide complementary information, it would not allow direct observation of substrate transport, which is the key advantage of using a single-molecule reporter. As further validation, liposome experiments using α -hemolysin pores (**New Supplementary Fig. 6e**) show that 90-95% of OppA molecules are capable of binding peptide and shifting to the high-FRET state, confirming the sensor's functionality.

In addition, we provide now evidence that single TmrAB transporters are reconstituted in the liposomes by labeling TmrAB with a single AF647 fluorophore and counting one-step and two-step photobleaching events (**New Supplementary Fig. 6a-c**), as noted in our response to Reviewer #1 (comment 13). This demonstrates that the majority of liposomes contain only one functional transporter, ruling out multiple transporter events.

Fig. R2: OppA exhibit a near-complete shift towards the high-FRET state after three sequential peptide translocation steps by TmrA^{EQB} (n = 192).

- 2) The legend for **Fig. 3f** on ‘transition kinetics’ is not relevant to TmrAB as the FRET is only reporting on OppA binding kinetics which is not the focus of this study. This should be changed.

Reply: We thank the reviewer for this comment. The figure legend of Fig. 3f has been updated to read: “Distinct OppA peptide-binding kinetics”. This change clarifies that the FRET signal

reports on OppA sensor binding rather than TmrAB conformational changes, ensuring accurate interpretation of the data.

- 3) **Fig. 2** would also benefit from having an HMM analysis table as Fig. 3, and make comparisons between the wt and mutant protein in the discussion.

Reply: We thank the reviewer for this suggestion. While HMM analysis is highly informative for TmrA^{EQB}, a direct comparison with TmrAB^{WT} is not feasible due to differences in experimental conditions:

(i) Temperature difference: TmrAB^{WT} transport was initiated on the microscope at 30 °C, whereas TmrA^{EQB} assays were conducted at 45 °C, which substantially affects transport activity (see also Reviewer #3, point 7).

(ii) Measurement setup differences: TmrAB^{WT} transport was recorded over time in 10-min intervals, limiting the number of observed molecules. In contrast, TmrA^{EQB} measurements were performed as 5 min end-point reactions, followed by 1 h imaging, enabling sufficient molecule counts for robust HMM analysis.

Given these difference, **Fig. 2** primarily demonstrates sensor's capability for TmrAB^{WT}, while HMM analysis is applicable and statistically meaningful only for TmrA^{EQB}.

- 4) Under the cryo-EM results section, the manuscript would benefit by providing a comparison of the NBDs with and without Mg²⁺; ie do the side chains adopt different rotamers? They should also provide rmsd values for the two structures. Are there any conformational changes within the NBD/coupling helix region that the absence of Mg²⁺ can induce?

Reply: We thank the reviewer for this insightful suggestion. We have now directly compared the NBDs of TmrA^{EQB} bound to ATP in the absence of Mg²⁺ with the previously reported Mg-ATP complex (PDB 6RAI; EMD-4776), as shown in **New Supplementary Fig. 14**. The RMSD between the two structures is 0.51 Å across all Cα atoms, indicating that Mg²⁺ removal has negligible impact on the overall global structure of the ATP-bound outward-facing occluded (OF^{occluded}) state of TmrA^{EQB}.

Nonetheless, subtle rearrangements are observed within the nucleotide-binding sites: residues coordinating Mg²⁺ (TmrA^{T400}, TmrB^{Q419}) adopt slightly different rotamers in its absence, as illustrated in **New Supplementary Fig. 14b–e**. No significant conformational changes are detected in the NBD-coupling helix regions, suggesting that Mg²⁺ removal does not substantially perturb the structural elements mediating NBD-TMD communication.

- 5) The statement in page 10 'Yet our work uncovers a previously overlooked Mg²⁺- independent ATP-bound state that supports single-turnover transport.' is confused with their previous sentence and there is not much relevancy for transporters which are greatly uncoupled for ATP-dependent substrate transport.

Reply: We thank the reviewer for this helpful comment. To avoid confusion and ensure the manuscript remains focused on the relevant mechanistic aspects to TmrAB transport, we have removed this sentence from the revised manuscript.

- 6) Supplementary Table 1 is reporting the EQ mutant structure but there is no figure of its ATP binding site as for the wt and any discussion beyond the difference maps. Supplementary Fig 8 should show the process of both wt and EQ mutant separately.

Reply: We thank the reviewer for this helpful comment. Views of the overall structure of TmrA^{EQB} and its nucleotide-binding sites are now included as **New Supplementary Fig. 14**. Additionally, the cryo-EM workflow for TmrA^{EQB} has been added as **New Supplementary Fig. 13**, allowing a direct comparison with the TmrAB^{WT} dataset. These additions provide a clearer visualization of structural similarities and differences between the WT and EQ mutant.

Reviewer #3 (Remarks to the Author)

Single-molecule dynamics reveal ATP binding alone powers substrate translocation by an ABC transporter.

The authors show that binding of ATP is sufficient for a half turnover of the peptide ABC transporter TmrAB, i.e. the transport of a peptide from cis to trans. Mg-ATP is required for hydrolysis and the other half turnover, which resets the system for the next translocation event. Thus, the conformational switch from inward-facing to outward-facing by ATP (without Mg²⁺ ions) is a key finding of this paper, which has been postulated before but may not have demonstrated as convincingly as done for TmrAB in this paper. Perhaps, the authors should summarize in their discussion to what extent the ATP-driven half turnover is unique for TmrAB (or heterodimeric ABCs), and indicate other studies where this has been shown directly or indirectly or ruled out. Now only computational studies and CFTR (special case) are mentioned in the Discussion section.

Overall, this is a beautifully executed single-molecule study on an important bacterial homologue of the human TAP system. The work is important and well presented. We have a few questions and minor comments for the authors to consider:

Reply: We sincerely thank Reviewer #3 for the thoughtful and encouraging evaluation of our work. We are delighted that the reviewer considers our single-molecule study to be “**beautifully executed**” and recognizes the **importance of TmrAB as a model for the human TAP system**. We greatly appreciate the acknowledgment that our findings **convincingly** demonstrate ATP-driven half-turnover and the associated conformational switch in TmrAB, underscoring the **novelty and mechanistic relevance** of our study.

In response to the reviewer’s helpful suggestion, we have expanded the Discussion section to include a broader comparison of ATP-driven half-turnover mechanisms across ABC transporters. Specifically, we now summarize previous biochemical and structural studies that report related phenomena in heterodimeric ABC exporters and highlight where such transitions have been directly or indirectly observed, or ruled out. These additions place our findings in a wider mechanistic context and clarify the extent to which the ATP-driven half-turnover observed in TmrAB may represent a general feature of asymmetric ABC transporters.

Specific Comments:

- 1) Figure 1d reports $K_D = 0.9 \mu\text{M}$ (text indicates values from 0.2-0.9 μM) for RRYQKSTEL binding to OppA; Figure 1f reports k_{on} and τ , from which the K_D can be calculated, which then yields $K_D = 7 \mu\text{M}$. What is the explanation for the approximately 10-fold difference in binding constant?

Reply: We thank the reviewer for raising this important point. The apparent difference between the K_D values arises from differences in experimental conditions and subsets of data analyzed.

Ensemble measurements (tryptophan quenching, $K_D = 0.22 \pm 0.04 \mu\text{M}$; ensemble FRET, $K_D = 0.3 \pm 0.1 \mu\text{M}$) report the thermodynamic equilibrium K_D of OppA for peptide RRYQKSTEL in solution. Single-molecule experiments with surface-immobilized OppA yield a slightly higher K_D ($0.9 \pm 0.2 \mu\text{M}$), likely reflecting minor effects of immobilization on peptide accessibility or protein conformational flexibility.

By contrast, Fig. 1f reports a kinetic K_D calculated from association (k_{on}) and dissociation (k_{off}) rates derived only from dynamic traces showing FRET transitions. Static molecules that remain in the high-FRET state are excluded, which can result in an overestimation of K_D . Additionally, peptide rebinding events may influence the observed kinetic rates.

Such differences between thermodynamic and kinetic K_D values are well-documented; for example, de Boer and colleagues observed that OppA binding to peptide RPPGFSPFR showed a thermodynamic K_D of $5 \pm 3 \mu\text{M}$ (ITC) versus a kinetic K_D of $14 \pm 5 \mu\text{M}$ (smFRET) (Boer et al., 2019).

Overall, the ~10-fold variation is consistent with expectations and reflects the intrinsic differences between ensemble equilibrium and single-molecules kinetic analyses.

- 2) The internal volume of the vesicles is calculated for a diameter of 160 nm, whereas the peak in the vesicle concentration is around 200 (Fig. 3g). Why is 160 nm chosen, and how does the distribution of vesicles sizes (from 100 to 500 nm) affect the analysis of the data (e.g. in the context of max 1 OppA and 1 TmrAB per vesicle, and both proteins need to be in the same vesicle)?

Reply: We thank the reviewer for this insightful question. In the section “Rationale of single-substrate sensors in liposomes”, a vesicle diameter of 160 nm was used as a representative example to illustrate the underlying calculation and concept. At this size, the internal volume corresponds to approximately 2 aL, resulting in an effective peptide concentration of about 1 μM for a single peptide molecule. This choice was made for conceptual clarity.

We agree that the liposome diameter strongly influences luminal peptide concentration. For example, vesicles of 100 nm, 200 nm, and 300 nm in diameter correspond to single-peptide concentrations of ~3.2 μM , 0.4 μM , and 0.1 μM , respectively. Larger vesicles (>300 nm) are more likely to contain multiple OppA molecules and are therefore excluded from analysis.

As noted in our response to Reviewer #1 (comment 12), although the vesicle population ranges from 100 to 300 nm, the majority of vesicles immobilized at the surface are ~100 nm in diameter as confirmed by **New Supplementary Fig. 11**. At this size, the luminal peptide concentration closely matches the theoretical estimate for one peptide per vesicle, assuming a single-substrate translocation event.

In addition, to directly assess transporter copy number per liposome, we labeled TmrAB^{WT} containing a single cysteine with AF647. Using the same reconstitution protocol, one-step and two-step photobleaching events were counted (**New Supplementary Fig. 6**). This analysis demonstrates that 95.6% of liposomes contain a single reconstituted transporter (see comment 13 to Reviewer #1).

Consequently, the majority of analyzed liposomes contain one OppA sensor and one TmrAB transporter, enabling accurate single-peptide transport measurements.

- 3) Fig. S4 shows that nanobody binding does not affect the activity of TmrAB. Since a high protein-to-lipid ratio is used, what is the evidence that the reported activity is actually transport and not peptide binding? The same applies to the data of SFig.6. It should be possible to discriminate binding from transport by comparing the number of peptides translocated with the number of TmrAB molecules in the membrane.

Reply: We thank the reviewer for raising this important point. In both the FACS transport assay (**Supplementary Fig. 7**) and the LC-MS transport assay (**Supplementary Fig. 9**), a clear **ATP-dependent peptide accumulation** is observed relative to the ADP control, confirming that the signal reflects active transport rather than surface binding. In the LC-MS assay, residual peptide

binding artifacts are further removed by a carbonate wash step (100 mM Na₂CO₃, pH 11.5), which efficiently removes peptides adhering to the liposome surface.

In the smFRET-based assay, transport is directly monitored via the luminal OppA sensor, which shows a distinct shift from low-FRET to high-FRET upon peptide translocation. If peptides were merely binding TmrAB without being transported, no FRET change would occur, as verified in the ADP control experiments (Fig. 2b and 3b). Additionally, a control experiment where peptide is added after the first translocation event in the absence of Mg-ATP (Supplementary Fig. 10f) shows no further FRET change, demonstrating that peptide movement into the lumen is strictly ATP-dependent.

Notably, the protein-to-lipid ratio in smFRET experiments was very low (1:10,000 w/w), meaning that, statistically, only one in ten liposomes contains a single TmrAB complex, minimizing the likelihood of multiple transporters per vesicle. By contrast, ensemble assays used higher ratios (1:20 w/w) to increase the overall transport signal.

Taken together, the combination of smFRET, LC-MS, and FACS assays provides robust evidence that the signals reflect true peptide transport rather than mere binding events.

- 4) Do non-hydrolysable ATP analogues also trigger the half turnover of TmrAB?

***Reply:** We thank the reviewer for this insightful question. To address it, we performed experiments using the non (very slow)-hydrolysable ATP analog AMP-PNP, and the results are now included as **New Supplementary Fig. 17**. These experiments demonstrate that AMP-PNP does not trigger half-turnover or peptide transport by TmrAB, indicating that ATP binding alone, rather than analogues, is required to drive a single translocation event.*

- 5) How was OppA encapsulated in the vesicle lumen? Page 23 only describes the membrane reconstitution of TmrAB.

***Reply:** We thank the reviewer for the comment. Although the manuscript primarily describes TmrAB membrane reconstitution, OppA was co-added to the detergent-destabilized liposomes during this process. The transient openings created in the destabilized liposomes allow OppA to be encapsulated in the vesicle lumen concurrently with TmrAB incorporation. This co-reconstitution strategy ensures that the luminal OppA sensor is present and functional for monitoring peptide transport.*

- 6) Fig. 2a and 2b show a high FRET peak, which is not seen in Fig. 3a and 3b. Is this endogenously bound peptide in this particular preparation of OppA?

***Reply:** We thank the reviewer for this observation. The high-FRET peak in Fig. 2a and 2b arises from endogenously bound ligands in the OppA preparation. Even in the absence of added peptide and Mg-ATP, a small fraction of OppA remains in a static high-FRET state. The extent of this proportion strongly depends on the protein purification protocol, particularly on the efficiency of removing endogenous ligands via partial unfolding and subsequent refolding with guanidine hydrochloride. Optimized purification, as employed for the data in Fig. 3, minimizes this high-FRET fraction, resulting in a predominantly low-FRET apo state suitable for transport assays.*

- 7) Figure 2c shows the time dependence of peptide translocation (0-10 and 11-20 min) for wildtype TmrAB, but this is not shown for the EQ mutant in Figure 3c or 3d. Shouldn't it be an order of magnitude slower?

Reply: We thank the reviewer for this helpful comment. Indeed, ensemble assays show that TmrA^{EQB} exhibits ~1000-fold lower transport activity than wild-type TmrAB. However, the dataset in **Fig. 2** (TmrAB^{WT}) and **Fig. 3** (TmrA^{EQB}) are not directly comparable due to differences in experimental conditions.

The wild-type TmrAB experiments (**Fig. 2**) were performed at 30 °C, where transport proceeds slowly enough for time-resolved smFRET imaging. At higher temperatures, wild-type transport becomes too rapid to resolve individual events. In contrast, the TmrA^{EQB} experiments (**Fig. 3**) were performed at 45 °C: pre-heated imaging buffer was applied to immobilized liposomes for 5 min, followed by washing to stop activity before imaging. This setup allows accumulation of sufficient single-turnover events accumulated for quantitative analysis despite the mutant's slow kinetics.

To directly address the reviewer's point, control experiments with TmrA^{EQB} at 30 °C (**Supplementary Fig. 10d, e**) show a larger low-FRET population and a reduced high-FRET population compared with the 45 °C dataset, confirming that transport by TmrA^{EQB} is substantially slower at lower temperatures.

- 8) Figure 6 suggests that 1 ATP is hydrolyzed per peptide translocated. Is there evidence for this contention; they also write on page 10 that wildtype TmrAB exhibits futile hydrolysis of ATP.

Reply: We thank the reviewer for raising this important point. The ATP-to-peptide coupling ratio indeed depends on the transporter variant and the biochemical conditions.

For TmrA^{EQB}, our data support an approximately 1:1 coupling ratio: one ATP hydrolyzed per peptide translocated. As shown in **Fig. 4f**, TmrA^{EQB} hydrolyzes ~0.25 ATP/min in the presence of MgCl₂, corresponding to ~1.2 ATP molecules during the 5-min transport window, which is consistent with the single-peptide translocation events observed by smFRET. This 1:1 ratio was also previously observed (Stefan et al, 2020).

For wild-type TmrAB, under MgCl₂ conditions, ATP hydrolysis occurs at ~1 ATP/s (**Fig. 4e**), reflecting substantial futile turnover, as is common for many ABC exporters (Bock et al, 2019). Under these conditions, the ATP/peptide coupling ratio is therefore not 1:1.

Under Mg²⁺-free (EDTA) conditions, ATP hydrolysis is reduced to background levels (**Fig. 4e**), yet single-peptide transport still occurs (**Supplementary Fig. 16c, e**). In this scenario, the effective coupling ratio approaches one ATP binding event per peptide translocated, consistent with the half-turnover mechanism described in the manuscript.

In summary, a near 1:1 ATP-to-peptide ratio is supported for the TmrA^{EQB} system and for TmrAB^{WT} under Mg²⁺-free conditions, whereas wild-type TmrAB under normal Mg²⁺ conditions exhibits substantial uncoupled ATP hydrolysis.

- 9) Page 28: It is not entirely clear how FRET traces were selected for the FRET distributions. Were traces without dynamics discarded? What were the criteria?

Reply: We thank the reviewer for this question. All smFRET traces were initially processed using DeepFRET, which assigns a per-trace confidence score based on a pretrained neural network classifier. Following the approach of Thomsen et al., we applied a confidence threshold of 80%, and only traces exceeding this value were retained for further analysis (Thomsen et al, 2020).

For the FRET histograms, we included both static and dynamics traces that passed this confidence threshold, ensuring that the histograms represent the full molecular population. For the HMM analysis, only dynamic traces were used, as these contain state transitions necessary for extracting dwell times and kinetic parameters. Static traces lack transitions and thus do not contribute meaningful kinetic information.

This two-step approach ensures that the FRET distributions capture the overall population while the kinetic analysis relies exclusively on mechanistically informative traces.

- 10) Fig. S3a and b: Why a double peak in the SEC profile for wildtype TmrAB, which is not seen for the EQ protein.

***Reply:** We thank the reviewer for this observation. SDS-PAGE analysis confirmed that both peaks in the SEC profile of TmrAB^{WT} correspond to properly assembled transport complexes. The first peak (F1) represents DDM micelles containing TmrAB dimers, whereas the second peak (F2) corresponds to micelles containing a single TmrAB complex. The double peak reflects a time-dependent equilibrium between singly and doubly occupied micelles.*

*To clarify, we repeated the SEC purification and updated the figure; the front peak (F1) is now significantly reduced. A statement explaining this behavior has been added to the legend of **Supplementary Fig. 5**.*

- 11) Fig. S7a: The high FRET state after one translocation event is similar to the high FRET state after two translocation events of Fig. 3d. Similarly, in Fig. S10e, the high/low FRET ratio is lower after translocation event 2 than after 1 (and even after 3 events it may still be lower than after 1). A brief description on variations between experiments and reproducibility or other explanation would be helpful.

***Reply:** We thank the reviewer for this observation. The slightly different high-FRET populations observed in previous **Supplementary Fig. S7a**, now **Supplementary Fig. 10a** and **Supplementary Fig. 16c** primarily reflect experimental variability and statistical sampling differences.*

***First, Supplementary Fig. 10a** was recorded in the presence of the ionophore valinomycin, which is necessary to dissipate the membrane potential, may subtly influence liposome integrity or local membrane properties, modestly affecting the OppA smFRET sensor (Gutiérrez-Pineda et al, 2021; Hussein & Sarles, 2025; Rose & Jenkins, 2006).*

*Second, sample size differences contribute to apparent variations. **Fig. 3** datasets included ~800 molecules per condition, yielding smooth, robust histograms, whereas the supplemental experiments contained fewer molecules (**Supplementary Fig. 10a**: n = 148; **Supplementary Fig. 16c**, translocation event 1: n = 158; translocation event 2: n = 90, translocation event 3: n = 246), naturally resulting in fluctuations in the high-/low-FRET populations.*

Importantly, despite these differences, all datasets consistently support the same mechanistic conclusion: under ATP-EDTA conditions, TmrAB does not perform additional peptide translocation events after the initial ATP-driven half-turnover. The observed variations reflect normal sampling variability rather than inconsistencies in transporter behavior.

- 12) The description of the legend of Fig. S10g (Hidden Markov Model) is ambiguous and needs some rewriting.

***Reply:** We thank the reviewer for this suggestion. The legend has been revised and the figure renumbered as **Supplementary Fig. 16e**. It now clearly states that Hidden Markov Model (HMM) analysis was applied exclusively to dynamic OppA traces exhibiting FRET transitions, while static traces without observable transitions were excluded from this analysis to extract dwell time accurately.*

References:

- Blanco M, Walter NG (2010) Analysis of complex single-molecule FRET time trajectories. *Methods Enzymol* 472: 153–178. doi:10.1016/S0076-6879(10)72011-5
- Bock C, Zollmann T, Lindt KA, Tampé R, Abele R (2019) Peptide translocation by the lysosomal ABC transporter TAPL is regulated by coupling efficiency and activation energy. *Sci Rep* 9: 11884. doi:10.1038/s41598-019-48343-6
- Ciftci D, Huysmans GHM, Wang X, He C, Terry D, Zhou Z, Fitzgerald G, Blanchard SC, Boudker O (2020) Single-molecule transport kinetics of a glutamate transporter homolog shows static disorder. *Sci Adv* 6: eaaz1949. doi:10.1126/sciadv.aaz1949
- de Boer M, Gouridis G, Vietrov R, Begg SL, Schuurman-Wolters GK, Husada F, Eleftheriadis N, Poolman B, McDevitt CA, Cordes T (2019) Conformational and dynamic plasticity in substrate-binding proteins underlies selective transport in ABC importers. *eLife* 8: 44652. doi:10.7554/eLife.44652
- Fitzgerald GA, Terry DS, Warren AL, Quick M, Javitch JA, Blanchard SC (2019) Quantifying secondary transport at single-molecule resolution. *Nature* 575: 528–534. doi:10.1038/s41586-019-1747-5
- Gutiérrez-Pineda E, Andreozzi P, Diamanti E, Anguiano R, Ziolo RF, Moya SE, Rodríguez-Presa MJ, C. C (2021) Effects of valinomycin doping on the electrical and structural properties of planar lipid bilayers supported on polyelectrolyte multilayers. *Bioelectrochemistry* 138: 107688.
- Holden SJ, Uphoff S, Hohlbein J, Yadin D, Le Reste L, Britton OJ, Kapanidis AN (2010) Defining the limits of single-molecule FRET resolution in TIRF microscopy. *Biophys J* 99: 3102–3111. doi:10.1016/j.bpj.2010.09.005
- Husada F, Bountra K, Tassis K, de Boer M, Romano M, Rebuffat S, Beis K, Cordes T (2018) Conformational dynamics of the ABC transporter McjD seen by single-molecule FRET. *EMBO J* 37 doi:10.15252/embj.2018100056
- Husada F, Gouridis G, Vietrov R, Schuurman-Wolters GK, Ploetz E, de Boer M, Poolman B, Cordes T (2015) Watching conformational dynamics of ABC transporters with single-molecule tools. *Biochem Soc Trans* 43: 1041–1047. doi:10.1042/BST20150140
- Hussein EA, Sarles SA (2025) Spontaneous depletion of valinomycin-mediated ion current in droplet interface bilayers. *Langmuir* 41: 26697–26704. doi:10.1021/acs.langmuir.5c03024
- Kosmidis E, Shuttle CG, Preobraschenski J, Ganzella M, Johnson PJ, Veshaguri S, Holmkvist J, Moller MP, Marantos O, Marcoline F, Grabe M, Pedersen JL, Jahn R, Stamou D (2022) Regulation of the mammalian-brain V-ATPase through ultraslow mode-switching. *Nature* 611: 827–834. doi:10.1038/s41586-022-05472-9
- Rose L, Jenkins ATA (2006) The effect of the ionophore valinomycin on biomimetic solid supported lipid DPPE/EPC membranes. *Bioelectrochemistry* 70: 387–393.
- Stefan E, Hofmann S, Tampé R (2020) A single power stroke by ATP binding drives substrate translocation in a heterodimeric ABC transporter. *eLife* 9: 55943. doi:10.7554/eLife.55943
- Thomsen J, Sletfjerding MB, Jensen SB, Stella S, Paul B, Malle MG, Montoya G, Petersen TC, Hatzakis NS (2020) DeepFRET, a software for rapid and automated single-molecule FRET data classification using deep learning. *eLife* 9: 60404. doi:10.7554/eLife.60404
- Wang L, Johnson ZL, Wasserman MR, Levring J, Chen J, Liu S (2020) Characterization of the kinetic cycle of an ABC transporter by single-molecule and cryo-EM analyses. *eLife* 9 doi:10.7554/eLife.56451

Response to Final Reviewer Comments:

Reply to Reviewer Response: I have checked both the comments of Reviewer 1 and the responses of the authors. Reviewer #1 has valid points. The authors answered the question partially satisfactorily, e.g. see points 9 and 12. The largest concern raised is in the heterogeneity of the vesicles and the limitations of current reconstitution methods to obtain unilamellar vesicles. In my opinion, the authors couldn't have done much better than they did. Reviewer 1 has a valid point that casts doubt on some of the conclusions in the article. I believe the main conclusion, which is reflected in the article's title, remains valid. Please see below for specific comments.

We sincerely thank the reviewer for the time, care, and thoughtful attention devoted to evaluating both Reviewer #1's comments and our responses. We truly appreciate the acknowledgement of the efforts we made to address the points raised, particularly regarding vesicle heterogeneity and reconstitution limitations. In response, we have added a short paragraph in the Discussion to explicitly acknowledge this limitation while clarifying how it is mitigated:

"...Finally, while liposome size control after reconstitution is inherently limited, the applied protocols reliably generate predominantly unilamellar vesicles (DOI:10.1038/nprot.2007.519). Importantly, the assay design—using very low transporter-to-lipid ratios and orientation-specific surface immobilization—selectively removes empty, oversized, and non-functional liposomes, resulting in a highly enriched population of function proteoliposomes with an apparent size of ~100 nm (Supplementary Fig. 11).

We hope this addition addresses the concern and further strengthens the clarity and rigor of our manuscript.

Reply to Reviewer Response Point 9 and 12:

Liposomes were prepared by thin-film hydration, followed by sonication, five freeze-thaw cycles, and extrusion through a 100 nm polycarbonate membrane prior to reconstitution. Freeze-thaw cycling is known to substantially reduce multi-lamellarity, yielding approximately 91% unilamellar vesicles, while extrusion further disrupts the remaining multilamellar structures, as demonstrated previously (DOI: 10.1021/acs.langmuir.8b04256; DOI: 10.1021/acs.langmuir.8b04256; DOI: 10.1016/s0378-5173(01)00721-9; DOI:10.1016/0378-5173(87)90139-6). In addition, protein reconstitution was performed following the well-established protocol by Geertsma *et al.* (2008), which is reported to produce unilamellar proteoliposomes with a low fraction of multilamellar vesicles (DOI: 10.1038/nprot.2007.519.).

We acknowledge that the multi-lamellarity is not directly probed by the fluorescence readout itself. However, the assay design strongly disfavors contributions from multilamellar vesicles. Following extrusion through pores <200 nm, only ~9% of bilamellar vesicles are expected to remain. Moreover, the experimental configuration requires that OppA be encapsulated within the same vesicle as TmrAB for transport activity to be detected. Multilamellar vesicles, incorrectly oriented transporters, or liposomes lacking functional encapsulation of OppA are therefore intrinsically excluded from the functional and single-molecule analyses. Additionally, non-tethered liposomes were removed by washing, further enriching for assay-competent vesicles.

Regarding liposome size, nanoparticle tracking analysis (NTA) was performed on samples directly after reconstitution, a method known to overestimate the contribution of larger particles and to broaden the apparent size distribution. In contrast, samples used for single-molecule FRET measurements underwent an additional centrifugation step (2,000 × g, 2 min) prior to surface immobilization. This step efficiently removes large and aggregated liposomes, explaining the narrower size distribution observed under single-molecule conditions. Furthermore, highly fluorescent aggregates (e.g., the bright spot visible in

Supplementary Fig. S11) were excluded from analysis based on DiD membrane staining. As a result, the liposome population analyzed at the surface is effectively enriched for ~100-nm vesicles.

Taken together, the photobleaching statistics, established liposome preparation protocols, and the intrinsic selectivity of the assay design collectively support the conclusion that the analyzed proteoliposomes predominantly contain a single transporter embedded in unilamellar vesicles.

Reviewer Response to Point 13: The authors have done the requested experiment. I believe that the majority of the vesicles with TmrAB have 1 transport protein per liposome, however it is hard to believe that the accuracy is 96.5%.

The reported value of 95.6% was calculated based on 2,088 individual photobleaching traces. Of these, 1,997 traces (95.6%) displayed a single-step photobleaching event, consistent with the presence of a single transporter per liposome, while only 91 traces (4.4%) showed a two-step photobleaching event, indicative of two transporters per liposome (Supplementary Fig.6). These results are in good agreement with the chosen reconstitution conditions, which were designed to yield, on average, one transporter per ten liposomes, resulting in a population dominated by empty liposomes and a low probability of multiple transporters per vesicle. We therefore conclude that the vast majority (>90%) of transporter-containing liposomes harbor only a single reconstituted transporter.

ABC transporters represent a large family of primary active membrane transporters, which are conserved across all domains of life. They utilize ATP binding and hydrolysis energy to transport diverse substrates against concentration gradients across cellular membranes. Their involvement in human diseases, including multidrug resistance and genetic disorders, makes understanding their molecular mechanisms relevant. ABC proteins are categorized based on their nucleotide-binding domains (NBDs) and transmembrane domains (TMDs). While structural studies provided detailed insights into structural aspects of the transport mechanisms, single-molecule techniques have begun to overcome limitations of ensemble methods, providing insights into transport dynamics and revealing novel mechanistic details of substrate translocation events.

In the submitted paper, a novel single-molecule platform combining FRET-based peptide sensing with conformationally selective nanobodies aims at direct observation of individual ABC transporter substrate translocation events. The studied system was TmrAB a heterodimeric ABC transporter that mediates peptide translocation across bacterial membranes, functioning as a structural and functional homolog of the human TAP complex. Using a slow-turnover TmrAB variant (TmrAEQB), the authors claim to observe discrete transport events at the single-transporter level. Their interpretation is that ATP binding alone, even without Mg²⁺, can drive substrate translocation through an IF-to-OF conformational switch. These findings support a mechanistic model where ATP binding induces NBD dimerization and conformational switching, with peptide translocation occurring concurrently, followed by Mg²⁺-dependent ATP hydrolysis for transporter resetting. The authors suggest that their work establishes a single-molecule framework for understanding ABC transporter mechanisms, though limitations include temperature constraints for wild-type TmrAB studies and sensor saturation after few uptake events.

I find the topic and the approach of the paper timely, innovative and very interesting and in conclusion worthwhile to publish. The study was in parts done carefully, yet various conclusions and the main claim of the paper are not supported by the supplied data and the modelling. I thus render the paper in principle suitable for publication in Nature Communications due to its novelty and ideas, subject to the following major revisions that require additional experiments/data and analysis:

- The paper lacks a summary of smFRET work on ABC transporters from various groups in which the major mechanistic conclusions should be summarized and the advances made (also in comparison to the current approach) are compared in a fair way
- Relevant references in which single-transporter assays were introduced are not mentioned (Stamou group and others); these should be integrated into the introduction and the framing of the paper
- The capabilities of the developed assay should be compared against other format (current ref. 26-27), which allow real-time recordings. What are the pros and cons of the approach introduced?

- Arising problem 1: the liposome size should be smaller so that the sensor range with K_d of 1 μM is most sensitive to changes of 1 to 2 peptides, for this a diameter of 150 nm would be ideal (see below)

V (m ³)	r (nm)	d (exp)	N=1	N=2	N=3	4 μM (blink)	6 μM (blink)
5,23333E-19	0,00000005	100	3,17E-06	6,35E-06	9,52E-06	1,260605333	1,890908
1,02214E-18	6,25E-08	125	1,62E-06	3,25E-06	4,87E-06	2,462119792	3,69317969
1,76625E-18	7,5E-08	150	9,40E-07	1,88E-06	2,82E-06	4,254543	6,3818145

- Problem 2: the liposome size distribution together with the averaging the authors do will not allow them to make any conclusion since the differences between liposomes of 200 nm and 350 nm has tremendous differences in the molecule number per vesicle (10 for 4 μM for 200 nm and 54 for 350 nm); in principle the best solution would be to calibrate the liposome size in each image using an intensity-based size measure (e.g., with a lipid dye) relating the intensity distribution to the size distribution.
- The author also do not provide another absolutely essential control, which is related to their approach. Since they have a label-free approach (from the viewpoint of the transporter) they have no readout to see how many transporters per liposome (with varying size) actually caused the change of sensor read-out. What is needed: label the transporter and reconstitute it at the same protein-to-lipid ratio as in the normal assays and count how many transporters are in the liposome to understand this effect.
- Another possibility would be to perform control experiments in which 1 or 2 peptides are reconstituted into the liposomes (without transporter) but with sensor to extract kinetics off/on.
- To say it clearly: currently the paper cannot support any claims and interpretations made further on the mechanistic side since the controls are lacking, and importantly: the calculations and assumptions on the ability of the assay to see single-peptide transport are not correct (and not provided in the manuscript, so it is difficult to see where the error comes from or if different assumptions were made).

Reviewer #1 (Remarks to the Author)

ABC transporters represent a large family of primary active membrane transporters, which are conserved across all domains of life. They utilize ATP binding and hydrolysis energy to transport diverse substrates against concentration gradients across cellular membranes. Their involvement in human diseases, including multidrug resistance and genetic disorders, makes understanding their molecular mechanisms relevant. ABC proteins are categorized based on their nucleotide-binding domains (NBDs) and transmembrane domains (TMDs). While structural studies provided detailed insights into structural aspects of the transport mechanisms, single-molecule techniques have begun to overcome limitations of ensemble methods, providing insights into transport dynamics and revealing novel mechanistic details of substrate translocation events. In the submitted paper, a novel single-molecule platform combining FRET-based peptide sensing with conformationally selective nanobodies aims at direct observation of individual ABC transporter substrate translocation events.

The studied system was TmrAB a heterodimeric ABC transporter that mediates peptide translocation across bacterial membranes, functioning as a structural and functional homolog of the human TAP complex. Using a slow-turnover TmrAB variant (TmrAEQB), the authors claim to observe discrete transport events at the single transporter level. Their interpretation is that ATP binding alone, even without Mg^{2+} , can drive substrate translocation through an IF-to-OF conformational switch. These findings support a mechanistic model where ATP binding induces NBD dimerization and conformational switching, with peptide translocation occurring concurrently, followed by Mg^{2+} - dependent ATP hydrolysis for transporter resetting. The authors suggest that their work establishes a single-molecule framework for understanding ABC transporter mechanisms, though limitations include temperature constraints for wild-type TmrAB studies and sensor saturation after few uptake events

I find the topic and the approach of the paper timely, innovative, and very interesting, and in conclusion worthwhile to publish. The study was in parts done carefully, yet various conclusions and the main claim of the paper are not supported by the supplied data and the modelling. I thus render the paper in principle suitable for publication in Nature Communications due to its novelty and ideas, subject to the following major revisions that require additional experiments/data and analysis:

Reply: We sincerely thank Reviewer #1 for the thoughtful and thorough evaluation of our manuscript. We are very grateful for the positive feedback regarding the timeliness,

innovation, and overall interest of our study. We are encouraged by the assessment that our work is, in principle, suitable for publication in Nature Communications due to its novelty and conceptual contribution. We also appreciate the constructive feedback pointing out areas where additional experiments, data, and analysis are needed. We have carefully addressed all concerns raised, which we believe considerably strengthen both the robustness and clarity of our mechanistic conclusions.

Reviewer Response: I have checked both the comments of Reviewer 1 and the responses of the authors. Reviewer #1 has valid points. The authors answered the question partially satisfactorily, E.g. see points 9 and 12. The largest concern raised is in the heterogeneity of the vesicles and the limitations of current reconstitution methods to obtain unilamellar vesicles. In my opinion, the authors couldn't have done much better than they did. Reviewer 1 has a valid point that casts doubt on some of the conclusions in the article. I believe the main conclusion, which is reflected in the article's title, remains valid. Please see below for specific comments.

Specific Comments: 1) The paper lacks a summary of smFRET work on ABC transporters from various groups in which the major mechanistic conclusions should be summarized and the advances made (also in comparison to the current approach) are compared in a fair way

Reply: We thank the reviewer for this important suggestion. We have now expanded the introduction to include a concise summary of previous smFRET studies on ABC transporter and their mechanistic insights. This addition also clarifies how our approach complements and advances beyond these earlier works. The New Paragraph (page 3) reads:

“In particular, single-molecule Förster resonance energy transfer (smFRET) studies confirmed an alternating-access mechanisms in ABC transporters; for example, the bacterial homodimeric exporter McjD requires both substrate and ATP to adopt the outward-facing conformation (Husada et al, 2018). Moreover, combining smFRET with single-particle cryo-EM has enabled visualization of the ABC transporter MRP1 under turnover conditions (Wang et al, 2020). Conformational changes in substrate-binding proteins (SBP) from bacterial ABC importers were investigated by smFRET (de Boer et al, 2019; Husada et al, 2015). The SBPs were repurposed as smFRET sensors to track individual translocation events in secondary active transporters, providing quantitative amino acid transport rates (Ciftci et al, 2020; Fitzgerald et al, 2019).“

This addition summarizes prior single-molecule approaches, acknowledges their contributions, and positions our work within the broader context of smFRET-based ABC transporter research.

Reviewer Response: Additional studies are cited and I feel the authors have done a good job. The same goes for their answer to point 2 below.

2.) Relevant references in which single-transporter assays were introduced are not mentioned (Stamou group and others); these should be integrated into the introduction and the framing of the paper.

Reply: We appreciate the reviewer pointing out this omission. Although the Stamou's work was already included in the original manuscript, it was not cited in the introduction. We now reference these key methodological contributions directly in the introduction: "Single-molecule techniques overcome ensemble averaging and provide high-resolution insights into transport dynamics (Blanco & Walter, 2010; Holden et al, 2010; Kosmidis et al, 2022). The work is also discussed in the Discussion (Kosmidis et al, 2022), where we compare it to the "resting" state observed in our system. This strengthens the framing and situates our study within the broader landscape of single-transporter methods

3.) The capabilities of the developed assay should be compared against other format (current ref. 26-27), which allow real-time recordings. What are the pros and cons of the approach introduced?

Reply: We thank the reviewer for this useful suggestion. We have expanded the Discussion to explicitly compare our assays with real-time assay formats (Ciftci et al, 2020; Fitzgerald et al, 2019), highlighting advantages and limitations. In brief, our assay offers several advantages: (i) Selective immobilization of uptake-competent transporters via a conformationally nonselective nanobody. (ii) Direct observation of peptide translocation through an ABC transporter at the single-molecule level. These features allow mechanistic dissection of transport steps that remain hidden in ensemble assays. The main limitations, now stated more explicitly, include: (i) Inability to perform single-molecule assays on wild-type TmrAB under fully physiological conditions (>45 °C) due to instrument and sensor thermostability constraints. (ii) Sensor saturation after a few uptake events, which currently precludes determining whether transport cycles proceed in bursts or continuously. This expanded discussion highlights the complementary nature of our assays relative to existing real-time formats and provides a balanced view of its strengths and constraints

Reviewer Response: The pros and cons of the approach are now are well described in the Discussion.

4.) Figure 1; please add supplementary data of traces of all conditions with at least 20-30 traces per condition.

Reply: We thank the reviewer for this suggestion. We have now added representative single-molecule FRET traces for all conditions in the New Supplementary Figs. 3 and 4, providing comprehensive visualization of the single-molecule behavior across all experimental conditions

Reviewer Response: Has been done and the traces look fine.

5.) Figure 1d; analysis of data is not correct or not correctly described. Normalized acceptor fluorescence is not identical to the area of the “blue” closed-state population shown in panels c; it would have to be the normalized area of one against the other population. According to panel c, there is saturation at closed-state populations of 60% so the panel needs attention

Reply: We appreciate the reviewer’s accurate observation. We have corrected the legend and text associated with Fig. 1d to clearly distinguish between normalized acceptor fluorescence and the population areas derived from Gaussian fits in Fig. 1c. This removes the previous ambiguity.

Reviewer Response: Has been corrected

6.) Figure 1c; please explain why the sensor has the limitation that only 60% are active or why the sensor does not saturate in the fully closed state? The problem persists in liposomes (Figure S2) but here also the apo state shows a substantial proportion of closed state in the absence of peptide. Maybe analysis of traces from individual liposomes can help? Please comment and explain.

Reply: We thank the reviewer for raising this important point. The apparent plateau at ~60% does not reflect intrinsic limitation of the sensor. Instead, it arises from constraints in Gaussian deconvolution when fitting small subpopulations in FRET histograms. As a result, the high-FRET fraction is underestimated, whereas inspection of the raw, higher-resolution histogram shows a larger proportion of molecules in the closed/high-FRET state. A similar underestimation was also observed in our initial liposome measurements. We have now clarified this in the manuscript and added a brief explanation in the figure legend. As outlined in our response to Reviewer #2 (comment 1), the histograms represent a mixture of low- and high-FRET OppA populations. Peptides are released after 0.5 s resulting in a shift from high to low FRET and the peptide concentration determines how fast the reassociation happens. A small low FRET population will be always present and achieving

complete sensor saturation requires multiple peptide translocation events and very high peptide concentrations. To directly demonstrate this, we performed an additional experiment in which three success translocation events mediated by TmrAEQB resulted in full sensor saturation (Fig. R2). Importantly, after optimizing OppA purification and sample handling, we were able to eliminate nearly all of the high-FRET apo population. The New Supplementary Fig. 17 (AMP-PNP control) confirms a clean low-FRET baseline in the absence of peptide. Finally, we note that our HMM analysis is based solely on dynamic, single-liposome trajectories and is not affected by these histogram-fitting artifacts.

Reviewer Response: Here we have both a complex question and a complex answer. SFig17 appears to support the authors arguments and is plausible.

7.) Figure 1e,f; please provide the dwell-time histograms of the different conditions used to calculate rates in panel f; to me the trace shown in e, which I assume is close to KD, should have a closed-state lifetime longer than 500 ms; more data/analysis is needed to give confidence in the analysis

Reply: We thank the reviewer for emphasizing the importance of dwell-time analyses. As noted, individual OppA molecules exhibit considerable heterogeneity in closed-state lifetimes, even within the same smFRET trajectory. Some closing events last only a few milliseconds, while others persist for seconds. This variability reflects the intrinsic stochastic dynamics of OppA and illustrates why single-molecule measurements are essential to resolve behaviors that are averaged out in ensemble analyses. The rates reported in Fig. 1f are based on a large dataset of transitions ($n = 1,311$), collected from multiple molecules across all conditions. Although individual traces – such as the example shown in Fig. 1e – may include events with closed-state lifetimes longer than 500 ms (or shorter), these fluctuations are fully incorporated into the overall statistical analysis. Comparable heterogeneity in OppA closed-state dwell times has been reported previously; for example, (de Boer et al, 2019) (Supplementary Fig. 2-1) observed variations of up to a factor of three within a single smFRET trace at peptide concentration near the KD. In response to the reviewer's request, we now provide dwell-time histograms for all experimental conditions used to derive the rates in Fig. 1f. These are included as Fig R1. The histograms summarize open- and closed-state dwell times under each experimental condition. We also note that temporal resolution is inherently limited by the acquisition speed of the microscope. Within this constraint, calculated closing rates show a clear linear dependence on peptide concentration. This relationship enables us to relate OppA closing kinetics to luminal peptide concentrations, which are in excellent agreement with theoretical estimates.

Reviewer Response: This is okay

8.) The above problems persist in Figures 2-3 and pose a challenge to interpret data in figure 3. How do the distributions of open/closed state of individual liposomes look like? If the main claim to make is that 1 or 2 peptides get transported than it would be good to analyze the data accordingly and show that in some liposome the on/off-rate or FRET histogram support this while in other maybe not transport has occurred; currently the interpretation and possibilities are averaged out in both figures and no data/analysis of individual liposomes is shown. I highly recommend to change this

Reply: We fully agree with the reviewer that single-liposome-level analysis is essential for a correct interpretation of the data in Figure 2. By calibrating liposome size and correlating the fluorescence intensity of individual liposomes with FRET sensor responses, we can directly monitor and quantify peptide transport events on a per-liposome basis. This refined analysis allows us to resolve discrete single-peptide transport steps and to distinguish liposomes that undergo one or two transport events from those that show no transport. Accordingly, the variability in open/close-state distributions across individual liposomes is now captured, rather than averaged out, and we show representative cases illustrating the expected heterogeneity

A detailed description of the reconstitution strategy, imaging workflow, and per-liposome analysis is provided in our response to point 12 (Reviewer #1) and in the revised Methods section. These additions clarify the homogeneity of the preparation and demonstrate that the observed transport events are robust at the single-liposome level

Reviewer Response: See below, point 12.

9.) The liposome size distribution is heterogenous with a population of 200 nm diameter (40%), but already this one has a substantial shoulder with 250 nm diameter (40%) and another large one >300-350 nm (20%); thus, any concentration calculation from rates of blinking will be completely different for these liposome size (up to factors of 5-10!)

Reply: We fully agree with the reviewer that liposome size heterogeneity can strongly influence concentration estimates. As detailed in our response to point 12 (Reviewer #1), we have now implemented improved liposome preparation and size characterization, followed by refined on-surface size calibration. These steps result in a substantially more homogeneous population of liposomes with diameters of ~100 nm diameter. This improved uniformity markedly reduces variation in the luminal volume and therefore minimizes differences in peptide concentration per liposome after transport

Reviewer Response: I find this answer not entirely satisfying. Why now suddenly an improved liposome preparation? What is different in the protocol, and where does one see that the improved method produces a more homogenous population? The vesicles will

remain heterogenous when produced by the used method. The (poly)dispersity of the vesicles should be presented, preferably by cryo-EM.

10.) The estimation made in panel f to relate 4 μM to 1 and 6 μM to 2 peptides is incorrect and thus fundamental conclusions made in the paper are not supported/valid. The paper says “These luminal peptide concentrations matched theoretical estimates for approximately one and two peptides in liposomes ranging of 200 ± 10 nm in diameter (Fig. 3g), ...”. My own estimate of the calculations in excel below shows a fundamental problem of the idea of the authors, i.e., to distinguish 1 or 2 peptide molecules but also use a heterogenous mixture of liposomes for this. In other words – currently it is not possible (explained below)

Reply: We acknowledge this limitation. To address it, we have added more detailed information on liposome reconstitution, centrifugation, and size calibration. This workflow produces a uniform population of ~ 100 nm diameter liposomes. With this well-defined size, the theoretical luminal concentration of a single peptide per liposome ($\sim 3.2 \mu\text{M}$) aligns with the sensor’s K_D value, allowing us to distinguish single- versus multiple-peptide transport events. These updated methods and corresponding data are presented in the New Supplementary Fig S11 and included in Fig. 3g

Reviewer Response: See response to point 9.

11.) Arising problem 1: the liposome size should be smaller so that the sensor range with K_D of 1 μM is most sensitive to changes of 1 to 2 peptides, for this a diameter of 150 nm would be ideal (see below)

Reply: Following this suggestion, we optimized our liposome preparation to achieve a mean diameter of ~ 100 nm. At this size, the luminal peptide concentration resulting from singlepeptide occupancy falls within the optimal detection range of the FRET sensor ($K_D \approx 1 \mu\text{M}$), thereby ensuring high sensitivity to single-peptide events while maintaining robust and reliable signal detection. This adjustment also minimizes variability due to liposome size heterogeneity.

12.) Problem 2: the liposome size distribution together with the averaging the authors do will not allow them to make any conclusion since the differences between liposomes of 200 nm and 350 nm has tremendous differences in the molecule number per vesicle (10 for 4 μM for 200 nm and 54 for 350 nm); in principle the best solution would be to calibrate the liposome size in each image using an intensity-based size measure (e.g., with a lipid dye) relating the intensity distribution to the size distribution.

Reply: We fully agree with this point and have implemented the following improvements: (i) Liposome size determination on the microscope surface: Liposomes were stained with the lipophilic dye DiD and immobilized on PEG-biotin surfaces via nanobody interactions to capture single TmrAB transporters in an uptake-competent orientation. Fluorescence intensity of individual liposomes was used to infer their diameters, confirming homogeneity and revealed that the majority of captured liposomes at the surface have a diameter of ~100 nm (New Fig. 3g and New Supplementary Fig. 11). (ii) Implications for peptide quantification: With this uniform liposome population, single-peptide concentrations correspond to ~3.2 μM , enabling accurate analysis of single-transport events and eliminating artifacts from averaging across heterogeneous liposome sizes. These improvements ensure that the single-liposome FRET data reliably supports our conclusions regarding single-peptide transport events.

Reviewer Response: I feel this point remains, even after calibration of liposome size. The problem is not only the accuracy of the calibration method but also the fact that a fraction of the liposomes is multilamellar. This is not captured by fluorescence, but clear from cryo-EM studies. Point 12 of Reviewer 1, and Point 2 of Reviewer 3 are very similar. The author response is adequate if indeed all vesicles have a diameter of ~100 nm and are unilamellar, which needs to be substantiated by data. In fact, Supplementary Figure 11d shows a main peak from 60 to 140 and a second peak from 150 to 200 nm.

13.) The author also do not provide another absolutely essential control, which is related to their approach. Since they have a label-free approach (from the viewpoint of the transporter) they have no readout to see how many transporters per liposome (with varying size) actually caused the change of sensor read-out. What is needed: label the transporter and reconstitute it at the same protein-to-lipid ratio as in the normal assays and count how many transporters are in the liposome to understand this effect

Reply: We appreciate the reviewer's suggestion. To address this, we carefully controlled the protein-to-lipid ratio during reconstitution such that, statistically, only one transporter is present per ~10 liposomes. Consequently, the majority of liposomes are empty and removed during washing, leaving a small population containing a reconstituted TmrAB. Additionally, to directly assess transporter copy number per liposome, we labeled TmrABWT containing a single cysteine with AF647. Using the same reconstitution protocol, one-step and two-step photobleaching events were counted (New Supplementary Fig. 6). This analysis demonstrates that 95.6% of liposomes contain a single reconstituted transporter. This stochastic reconstitution approach, combined with orientation-selective

immobilization, ensures that the observed FRET responses predominantly arise from single transporters per liposome.

Reviewer Response: The authors have done the requested experiment. I believe that the majority of the vesicles with TmrAB have 1 transport protein per liposome, however it is hard to believe that the accuracy is 96.5%.

14.) Another possibility would be to perform control experiments in which 1 or 2 peptides are reconstituted into the liposomes (without transporter) but with sensor to extract kinetics off/on.

Reply: We thank the reviewer for this suggestion. However, due to the stochastic nature of encapsulation, it is not feasible to prepare liposome populations containing exactly one or two peptides without introducing substantial heterogeneity. In contrast, our TmrAEQB single-turnover assay inherently restricts peptide transport to single events per liposome within the assay window, providing a well-defined system to accurately quantify single-peptide translocation

Reviewer Response: I agree with the authors

15.) To say it clearly: currently the paper cannot support any claims and interpretations made further on the mechanistic side since the controls are lacking, and importantly: the calculations and assumptions on the ability of the assay to see single-peptide transport are not correct (and not provided in the manuscript, so it is difficult to see where the error comes from or if different assumptions were made).

Reply: We thank the reviewer for this important comment. To address these concerns, we have included extensive additional controls, expanding the Supplementary Information from Fig. 10 to Fig. 17. Mechanistic interpretations based on single-molecule results are further corroborated by orthogonal approaches, including cryo-EM, radioactivity-based ATPase measurements, and ensemble transport assays based on LC-MS. Regarding the theoretical calculations of single-peptide encapsulation, the original calculations were correct. To enhance clarity and transparency, we have now explicitly provided these calculations in the Methods section. With the updated, well-defined size of surface-immobilized liposomes (~100 nm diameter), the theoretical luminal concentration of a single peptide aligns closely with the observed smFRET data. These improvements confirm that our experimental setup reliably reports single-peptide transport events, thereby supporting the mechanistic conclusions presented in the manuscript.

Reviewer Response: The reviewer's request for controls is not very clear, but I agree with the authors that a substantial amount of extra work has been included in the revised manuscript. I do not agree that the paper cannot support any claims on the mechanism.

However, I would tone down the claims on the uniformity of vesicles, which in my view are not supported by irrefutable data.